

# Multiresolution analysis of the spatiotemporal variability in global radiation observed by a dense network of 99 pyranometers during the HOPE campaign

Bomidi Lakshmi Madhavan, Hartwig Deneke, Jonas Witthuhn, and Andreas Macke

Leibniz-Institute for Tropospheric Research (TROPOS), Permoserstraße 15, 04318 Leipzig, Germany

*Correspondence to:* B. L. Madhavan (madhavan.bomidi@tropos.de)

**Abstract.** The time series of global radiation observed by a dense network of 99 autonomous pyranometers are investigated with a multiresolution analysis based on the maximum overlap discrete wavelet transform and the Haar wavelet. For different sky conditions, typical wavelet power spectra are calculated to quantify the timescale dependence of variability in global transmittance. The power spectra of global transmittance are found to be dominated by the direct irradiance component under all sky conditions. Distinctly higher variability is observed at all frequencies in the power spectra of global transmittance under broken cloud conditions compared to clear, cirrus or overcast skies. The spatial autocorrelation function including its frequency-dependence is determined to quantify the degree of similarity of two time series measurements as a function of their spatial separation. Distances ranging from $100\,\text{m}$ to $10\,\text{km}$ are considered, and a rapid decrease of the autocorrelation function is found with increasing frequency and distance. For frequencies below $1.0\,\text{min}^{-1}$, variations in transmittance become completely uncorrelated already after several hundred meters. A method is introduced to estimate the deviation between a point measurement and a spatially averaged value for a surrounding domain, which takes into account domain size and averaging period, and is used to explore the representativeness of a single pyranometer observation for its surrounding region. Two distinct mechanisms are identified, which limit the representativeness: on the one hand, spatial averaging reduces variability and thus modifies the shape of the power spectrum. On the other hand, the correlation of variations of the spatially averaged field and a point measurement decreases rapidly with increasing temporal frequency. For a grid-box of $10\times10\,\text{km}^2$ and averaging periods of $1.5$–$3\,\text{h}$, the deviation of global transmittance between a point measurement and an area-averaged value depends on the prevailing sky conditions: 2.8% (clear), 1.8% (cirrus), 1.5% (overcast) and 4.2% (broken clouds). The global radiation observed at a single station is found to deviate from the spatial average by as much as $14$–$23\,\text{W}\,\text{m}^{-2}$ (clear), $8$–$26\,\text{W}\,\text{m}^{-2}$ (cirrus), $4$–$23\,\text{W}\,\text{m}^{-2}$ (overcast), and $31$–$79\,\text{W}\,\text{m}^{-2}$ (broken clouds) from domain averages ranging from $1\times1\,\text{km}^2$ to $10\times10\,\text{km}^2$ in area.

## 1 Introduction

The sun is the primary source of energy for the Earth's climate system. Clouds strongly modulate the radiation budget through reflection of solar radiation back to space, and by trapping terrestrial radiation within the atmosphere (Trenberth et al., 2009). A better understanding of the small-scale variability in the radiation field at the surface resulting from clouds will have numerous





practical applications, ranging from climate related research focused on cloud radiative effects and cloud-aerosol interactions, to the representation of radiative transfer in numerical weather prediction (van den Hurk et al., 1997) and to solar energy forecasting (Robles Gil, 2007). According to the latest Intergovernmental Panel on Climate Change report, the impact of various cloud types on the net radiation budget is not fully understood to the extent that for some cloud types neither the

magnitude nor even the sign is known (Boucher et al., 2013).

This can be attributed to our currently still very limited understanding of cloud processes and the resulting cloud-radiation interactions, due to their complexity and the wide range of scales involved. Small-scale processes such as up- and downdraughts, turbulent mixing, together with the availability and composition of cloud condensation nuclei and large-scale dynamics influence the formation and life cycle of clouds, which subsequently determine their optical properties and thus their

interaction with radiation (e.g. Baker, 1997; Scheirer and Macke, 2003; Baker and Peter, 2008). In consequence, clouds induce the largest amount of uncertainty in climate projections (Baker and Peter, 2008) and weather prediction (Stensrud, 2009).

Satellite observations are one very important source of information for investigating clouds and their radiative effects. Current operational retrievals of cloud properties from passive satellite sensors do however invoke the assumption of plane-parallel, horizontally homogeneous clouds. While these retrievals have been extensively evaluated with ground-based

measurements over the past years (e.g. Min et al., 2003; Roebeling et al., 2008; Madhavan et al, 2012), significant biases and uncertainties remain due to the limitations of this assumption (e.g. Horvath et al., 2014).

These complications can be mainly attributed to horizontal photon transport, radiative smoothing and sub-pixel inhomogeneity (e.g. Cahalan et al., 1994; Barker and Li, 1997). To address these issues, Pincus et al. (1999) proposed a parameterization to account for unresolved sub-grid scale variability, which does however depend on a priori information

about typical variability for different cloud types. They also identified an increase in optical thickness and a decrease in relative variability in the transition from cumuliform to stratiform clouds. Oreopoulos et al. (2000) studied power spectra obtained from high-resolution Landsat observations, and identified different behavior for scales below 1 km and within the interval from 1 to 5 km as a consequence of both cloud morphology and 3D cloud radiative effects. Based on a large ensemble of 3D cloud fields as input for 3D radiative transfer models, Schewski and Macke (2003) could show that especially transmitted solar radiation

and domain average cloud properties are highly correlated. In related research, Venema et al. (2006) and Schmidt et al. (2007) showed that a stochastic cloud generator together with a 3D radiative transfer model can be used to link the statistical properties of cloud observations to those of the resulting solar radiation field with satisfactory accuracy.

The attribution of deviations between ground-based observations, satellite observations and model results is also complicated by the effects of spatial collocation and the limited representativeness of a point measurement for domain averages implicitly

assumed in any such comparisons (e.g. Deneke et al., 2009; Schutgens and Roebeling, 2009; Greuell and Roebeling, 2009). Large inconsistencies are expected to occur in particular for short time periods ($< 1\,\mathrm{h}$) and broken cloud fields, if point measurements are compared to large satellite pixels or coarse-resolution model output ($> 1\,\mathrm{km}$).

Focusing on solar radiation, Nunez et al. (2005) concluded that for stratocumulus clouds, a high frequency of observations is required to estimate the hourly-averaged global radiation from satellite with acceptable accuracy ($\sim 5\,\%$ error for six scans

per hour). To estimate the representativeness of a point measurement for a larger domain, Long and Ackerman (1995) used



data from a network of pyranometers during the First ISCCP (International Satellite Cloud Climatology Project) Regional Experiment (FIRE), and showed that a spatial separation between the measurement sites of up to 150 km can be allowed for daily averages. This is, however, mainly attributable to the fact that correlation is dominated by the diurnal cycle of solar radiation at the top-of-atmosphere. In another study, Barnett et al. (1998) found a characteristic timescale of 60 min for solar

radiation on cloudless days, and twice that long for cloudy days, after removal of the diurnal cycle component. They also conclude that to achieve a correlation of 0.9 between measurements at a point and averaged over a surrounding area on cloudy days, the central site can be considered representative for a region with a radius of 30 km. Both Barnett et al. (1998) and Duchon and O'Malley (1999) report that the representativeness of a point measurement for area averages depends on the considered averaging time and the prevailing cloud type.

Comparing satellite-based solar radiation retrievals from the Advanced Very High Resolution Radiometer (AVHRR) to pyranometer observations, Deneke et al. (2005) report a large root mean square error ($rmse$) of $86\,\mathrm{W\,m^{-2}}$ for individual station records even when averaging over $10 \times 10$ satellite pixels and over 40 min. In contrast, a much better accuracy ($rmse \sim 33\,\mathrm{W\,m^{-2}}$) is achieved if the average of 30 stations is considered. They interpret this finding as evidence that a significant fraction of the $rmse$ in the comparison results from the variability of the global radiation field due to the limited

representativeness of the pyranometer measurements for the satellite-retrieved values.

Over the past decades, several ground-based surface radiation networks have been established (e.g. Barker et al., 1998; Ohmura et al., 1998; Michalsky et al., 1999). However, a dense network of solar radiation measurements at the surface with station distances smaller than a typical satellite pixel or model grid has to our knowledge not been realized before. Such a network has been developed and operated during the **H**igh Definition Clouds and Precipitation for advancing Climate Prediction

(HD(CP)$^2$) **O**bservation **P**rototype **E**xperiment (HOPE) conducted around Jülich, Germany (Madhavan et al., 2016). This unique data set can provide insights into the small-scale variability of global radiation due to various cloud types, and possibly enable the development of parametrizations of the unresolved spatio-temporal variability in the radiation field. Using this data set, Lohmann et al. (2016) explored the fluctuations of the clear-sky index (i.e., the ratio of instantaneous global radiation to the radiation on the Earth with a cloud-free atmosphere) on clear, overcast and mixed sky conditions with a simple increment

statistics to study the smoothing effects of distributed photovoltaic power production.

Spatial and temporal scaling properties of the time series of observed global radiation can be derived using a wavelet-based multiresolution analysis. Wavelet-based estimators of variance, covariance and cross-correlation decompose their scale-independent counterparts on a scale-by-scale basis. Multiple studies have adapted similar wavelet-based methods to explore a wide range of subjects involving the atmospheric time series applications (Whitcher et al., 2000), solar radiation (Deneke et al.,

2009), fluctuation analysis of the power generated by photo-voltaic plants (Perpiñán et al., 2013), geophysical seismic signal analysis (Grosmann and Morlet, 1984), signal and image processing, and vegetation monitoring.

In our study, the statistical properties inferred from a multi-resolution analysis (MRA) of the time series of global radiation are subsequently used to quantify the representativeness of a point measurement for a surrounding domain considering typical domain sizes and different sky conditions. Instead of directly considering the global radiation, its transmission by the





atmosphere, denoted as global transmittance in this paper, is considered, as changes in incoming solar radiation are removed at least to first order. The present study is focused at addressing the following research questions:

    i. How do the power spectra of global transmittance differ for different sky conditions?

    ii. How representative is the time series observed at one station for other near-by stations?

    iii. How representative is the single station observation for domain averages considering different spatial and temporal averaging scales?

This paper is organized as follows: in Section 2, details of the observational data used in this study are presented. An overview of our methods is given in Section 3, with more details on the theory given in the Appendix. Section 4 discusses the results of the multiresolution analysis, the behavior of the power spectra, and the spatial correlation under different prevailing sky conditions. These results are further used to investigate the spatial representativeness of a point measurement for spatial averages over typical domain sizes, and to quantify the expected deviations. Finally, summary and conclusions with an outlook are presented in Section 5.

## 2   Data sets

As part of the HOPE campaign, a high density network of 99 autonomous pyranometer stations was operated across a spatial domain covering 50.85–50.95° N and 6.36–6.50° E ($\sim 10 \times 12\,\mathrm{km}^2$ area) around Jülich, Germany from 2 April to 24 July 2013. Each of these stations continuously recorded the global radiation ($G$ in $\mathrm{W\,m}^{-2}$) using a silicon photodiode pyranometer (Model: EKO ML-020VM) with 10 Hz resolution. A GPS (Global Positioning System) module embedded on the data acquisition board of each station provides an accurate time reference. The global radiation measurements have been averaged into 1 s time periods during the conversion of the ASCII log files into NetCDF data files following the Climate and Forecast Metadata Conventions version 1.6 (Eaton et al., 2011). From these measurements we have derived the global transmittance ($T$), which is calculated by normalizing the global radiation ($G$) under all sky conditions by the extraterrestrial radiation at the top of atmosphere assuming a value of the solar constant of $1360.8\,\mathrm{W\,m}^{-2}$ from Kopp and Lean (2011) and accounting for the cosine of the solar zenith angle and Sun-Earth distance. The solar zenith angle and the Sun-Earth distance have been calculated following the guidelines of WMO (2008).

The spectral range (0.3–1.1μm) of silicon sensors is a well-known limitation of this type of pyranometer due to the narrow spectral response of the photodiode. While the derived global transmittance is sensitive to aerosols and cloud optical thickness, information on cloud thermodynamics phase and cloud droplet effective radius is beyond the spectral range of these silicon photodiode pyranometers. As the spectral composition of the measured global radiation in the field deviates due to non-uniform spectral sensitivity, a measurement uncertainty of 5 % can be expected for higher solar zenith angles. Detailed information about the pyranometer network setup during the HOPE campaign, data processing, quality control and uncertainty assessment due to various potential sources of error are presented in Madhavan et al. (2016).



The real-time sky conditions were assessed using hemispheric images from a Total Sky Imager (TSI) operated at the Research Center Jülich (FZJ) during the HOPE campaign. Time-azimuth ($t - azi$) plots were generated from the TSI images. Every line in these $t - azi$ plots contains pixels from the azimuth angle range from $0°$ to $360°$, sampled at an elevation angle of $45°$. These plots capture both spatial and temporal variability of clouds, and help to identify the dominating advection direction of clouds, which shows up by sine like patterns (Löhnert et al., 2014). Since the ground-based observations have a field of view which does not exceed $50\,\mathrm{km}$ in radius (Henderson-Sellers et al., 1987), Meteosat SEVIRI (Spinning Enhanced Visible and Infrared Imager) images based on the day natural RGB color composites (Lensky and Rosenfeld, 2008) were additionally used for the physical interpretation and thermodynamic phase identification of the cloud types present over the observation domain. The 0.6, 0.8 and $1.6\,\mathrm{\mu m}$ spectral channels where enhanced in resolution using the high frequency component of the broadband HRV channel ($0.4$–$1.1\,\mathrm{\mu m}$)(Deneke et al., 2010). Based on the predominant sky conditions during the daylight period (6-18 h local time), we have classified selected days as clear, cirrus, overcast, or broken cloudy conditions (see Table 1).

As there were no simultaneous measurements of the direct and diffuse irradiance components during the HOPE campaign at Jülich, these were obtained from measurements by the MObile RaDiation ObseRvatory (MORDOR) station operated in parallel to the pyranometer network during the Melpitz-Column (MCOL) experiment (May – July 2015) around Melpitz, Germany, as high-quality reference measurements. The MORDOR system consists of a sun tracker with EKO instruments such as pyrheliometer (MS-56) to observe the direct irradiance, and two pairs of pyranometers (ML-020VM) operating in shaded and unshaded conditions to observe the diffuse and global radiation respectively. To fulfil the Baseline Surface Radiation Network (BSRN) quality requirements, secondary standard pyranometers (Kipp&Zonen CMP-21) are employed (McArthur, 2005). A second set of pyranometers of identical type EKO ML-020VM to the pyranometer network are also used. This allows to assess the absolute accuracy of the radiation network, and to resolve the high-frequency variations in radiation at frequencies above the time response of classical thermopile pyranometers. Additionally, the broadband longwave irradiance is observed with a pyrgeometer (Kipp&Zonen CGR-4). Hemispheric sky images are acquired by a fish-eye camera (Vivotek, FE8172v) to provide the real-time cloud type information. The measurements of diffuse and global radiation from the EKO pyranometers of MORDOR are used here to obtain the respective radiation components, to minimize instrumental differences. A more thorough comparison of MORDOR measurements and the pyranometer network is planned for the future.

## 3 Methods

### 3.1 Multiresolution analysis (MRA)

A multiresolution analysis (MRA) based on the maximum overlap discrete wavelet transform (MODWT) (Percival, 1995) and the Haar wavelet (Haar, 1910) is applied to the time series of global flux transmittance measurements of the pyranometer network. The Haar wavelet filters correspond to rectangular scaling and wavelet functions, which act as low-pass and bandpass filters respectively (see Figure 1). Maximum time localization is achieved through the minimal support of the filters. This also minimizes the range of edge effects. The choice of a rectangular function as low-pass filter has also the advantage that it corresponds to an arithmetic average for a specific period and is thus simpler to interpret than the weighted averages obtained





by other wavelets. The drawback of the rectangular function as a low-pass filter is its sub-optimal frequency separation, which could result in lower correlations found between time series than those obtained by Gaussian averaging. A summary of the methodology is given here, while a more formal mathematical treatment with relevant references to the literature can be found in the Appendix of Deneke et al. (2009).

In the MRA, the day is chosen as fundamental frequency $f = 1\,\mathrm{day}^{-1}$, and the frequency domain is partitioned into bands delimited by the harmonics $f_J$ given by $f_J = 2^J \cdot f$. For obtaining this partitioning, the original data set has been resampled from 86400 to $2^{16}$ (= 65536) samples per day before subjecting it to the MRA. To avoid aliasing effects caused by the resampling step, a 3 s running mean has been applied as low-pass filter prior to subsequent decimation of samples by a factor of $\frac{512}{675}$. Only harmonics from $J = 3$ to $J = 14$ with corresponding averaging frequencies of 3 h (= $2^{13}$ samples) and 5.25 s (= $2^2$

samples) are considered in the following analyses, to avoid the influence of changes in solar zenith angle below 75° and the anti-aliasing filter above this frequency range, respectively. The running means of the original time series for the different harmonics $J$ (corresponding to an averaging time period of $2^{-J} \times 86400\,\mathrm{s}$) are referred to as *smooths* denoted by $S_J$. Further, the differences between two subsequent smooths are called the *details*, $D_J (= S_{J+1} - S_J)$, and contain the variability (or fluctuation behaviour) within a frequency band delimited by two harmonics.

Figure 2 shows the results of the MRA applied to the global radiation ($G$) and to the global transmittance ($T$) for measurements from the pyranometer station located at FZJ (hereafter, referred as PYR76) on 25 April 2013 for scales $J = 3$ to $J = 12$ (i.e., 3 h to 21 s). On this day, light fog prevailed in the morning with some cirrus clouds. Thereafter, broken cumulus mediocris clouds were observed until late afternoon, followed by rapidly increasing low stratus clouds leading to an overcast sky by evening. The left panel of Figure 2 contains ten smoothed versions ($S_J, J \in [3, 12]$), the *smooths* of the original time

series corresponding to averaging time scales from 3 h to 21 s. The right panels of Figure 2 show the corresponding *details* ($D_J, J \in [3, 11]$). As the scale $J$ increases, the time series of transmittance details exhibit significant variability. Note that the MRA results were limited to solar zenith angles less than 75° to exclude edge effects. Based on Percival (1995), the maximum overlap discrete wavelet transform decomposes the variance of a time series on a scale-by-scale basis and can be estimated from the variance ($var$) of the MODWT coefficients as given below:

$$var(T) = var(S_J) + \sum_{j=1}^{J} var(D_j) \tag{1}$$

This result can be generalized to the calculation of the correlation, where the wavelet coefficients of two time series can be used to provide an estimate of the correlation at a given scale (Whitcher et al., 2000).

An effective graphical technique to the MRA is the *horizon graph* (Heer et al., 2009). As illustrative examples, the horizon graphs of the global transmittance details for different scales (see table 2) from PYR76 station is shown in Figure 3 for days with

different sky conditions: clear (4 May 2013), cirrus (16 July 2013), overcast (9 June 2013) and broken clouds (25 April 2013). Each row in the top panel of Figure 3 includes a different detail of the MRA, while the middle panel shows the original time series of global transmittance. In addition, the $t - azi$ plots at 45° solar elevation angle are included as lower panels to illustrate the sky conditions during each observation day. While the fluctuations in the transmittance at different lower frequencies can





be perceived from the contrasting color bands, significant variability can be observed in the situations with broken clouds even at high frequencies corresponding to periods of 1 min or shorter.

## 3.2 Spatial representativeness of point measurements

From the MRA, the wavelet power spectrum of transmittance can be calculated (Sec. 3.1), which describes the partitioning of
signal power into frequency ranges, and reflects the characteristics of the prevailing sky conditions. Additionally, the spatial autocorrelation function describes the similarity of variations in the time series measured at two stations as a function of their distance. By determining both the power spectrum and the frequency-dependent spatial autocorrelation function across the observation domain under different sky conditions, the representativeness of a point measurement for an area-averaged value can be quantified, including the expected deviation. Various statistical parameters, namely the variance, covariance and
explained variance linking the time series of a point measurement to that of an area-averaged value are derived in Appendix A. In this study, we consider three typical spatial areas ($A$) of interest with $1{\times}1\,\mathrm{km}^2$, $3.163{\times}3.163\,\mathrm{km}^2$ and $10{\times}10\,\mathrm{km}^2$. The expected deviation ($\delta$) between a point measurement and an area-averaged value for a surrounding domain is calculated as:

$$\delta = \sqrt{(1-\gamma_{S,J}^2)\cdot var(S_J) + \sum_{j=1}^{J}(1-\gamma_{D,j}^2)\cdot \alpha_{A,j}\cdot var(D_j),} \tag{2}$$

where the variance of the transmittance smooths ($S_J$) and details ($D_j$) are obtained from the power spectrum of the point
measurement, and $\alpha_A$ is a linear reduction factor relating the variance of the point measurement to the variance of an area-averaged time series. The explained variance (i.e., $\gamma_{S,J}^2$ and $\gamma_{D,J}^2$) between the point and area-averaged values are obtained separately for transmittance smooths and details for the different spatial and temporal scales. Then, the expected deviation for each wavelet detail is calculated based on the explained variance and summed to yield an estimate of the total variance, accounting for a reduced temporal variability of the spatially-averaged transmittance by the reduction factor.
Further, the estimated representativeness error of the transmittance ($\delta T$) time series can be converted into a deviation in global radiation ($\delta G$) by multiplication with a fixed value ofthe top-of-atmosphere solar irradiance, which avoids the known influence of changes in solar zenith angle. A fixed value of $680.4\,\mathrm{W\,m}^{-2}$ is used here, which is half the solar constant and is taken as an estimate of the daytime mean value during summer months for the considered region. This procedure can be adopted to improve photovoltaic power forecasting models under different sky conditions, especially with broken clouds, which
require absolute values of radiation instead of transmittance.

## 4 Results and Discussion

### 4.1 Power Spectra of Global Radiation

Wavelet-based power spectra characterize the variability contained in specific frequency intervals for both stationary and non-stationary processes by the spectral power density. As the time series of global transmittance results from a non-stationary
process (i.e. its statistical properties are not time-invariant), the wavelet power spectrum is a suitable tool for the analysis of





the variability contained within specific frequency intervals, and to study the effect of temporal and spatial averaging on the variability of the time series.

Power spectra of the global, direct and diffuse transmittance are shown in Figure 4. It is evident from the plots that the spectra for the direct and global components are very similar for all sky conditions. The spectral power density resulting from

variations of the diffuse transmittance is lower and only contains significant variations at low frequencies, again a conclusion valid for all sky conditions. A plausible explanation is the hemispherical field of view of the diffuse radiation observation, which is less sensitive to small-scale variations in cloud properties than the direct beam of sunlight.

This also implies that the variability in global transmittance is dominated by changes in direct transmittance at all frequencies. Boers et al. (2000) also demonstrated that the global radiation is very sensitive to cloud inhomogeneities, in particular for

broken cloud fields due to contributions from the direct radiation.

In Figure 5, the wavelet power spectrum of the global transmittance is shown together with the cumulative variance (or cumulative standard deviation, from Eq. 1) for different sky conditions, obtained as average power spectrum obtained from all pyranometer stations. The cumulative variance quantifies the fraction of variance resolved by an observation which has been smoothed with a specific averaging period, and is determined using the spectral power density decomposition given in Eq. 1. It

thus gives an indication for how much variability is lost if averaging is applied to the time series. As the frequency increases, the variability in global transmittance decreases, irrespective of the prevailing sky conditions. However, there are clear differences in the shapes of the power spectra for the different sky conditions. During situations with broken clouds, the variability of transmittance is distinctly higher than for all other cases, irrespective of the considered frequency interval. It is well-known that in the presence of broken clouds, multiple reflections and scattering events off the sides of clouds and at the surface lead

to significant horizontal photon transport and strong 3D radiative effects. For the associated types of low-level clouds, such as fair weather cumulus or towering cumulus, a high global transmittance can frequently be observed at the surface exceeding that of a clear sky (Schade et al., 2007). Similar effects also occur when patches of cirrus or alto-cumulus clouds are present in the field of view, but do not obscure the sun. Overall, for broken cloud fields, strong spatial and temporal variations are present over a wide range of frequencies.

On days with cirrus clouds, the spectral power density is lower than for broken clouds and higher than for clear skies. Interestingly, the variance of transmittance under an overcast sky is found to be lowest at high frequencies (i.e., 10.5—5.25 s), with a steep increase up to a time period of 11.25–22.5 min. Thereafter, the variability is slightly higher and comparable to that observed for cirrus cloud situations. Under a homogeneous overcast sky with optically thick clouds, the global radiation is contained completely in the diffuse component, and the radiance at the cloud base observed from the ground will be relatively

uniform over time. However, under partly overcast skies, the global transmittance of clouds is also influenced by multiple reflections of solar radiation between the surface and the cloud base, which causes an increased variance. Note that variations of the transmittance lower than the measurement uncertainty ($\pm 0.0013$) of our pyranometer stations are neglected here, which is the case for higher frequencies corresponding to time periods below 42 s.

The slope of the power spectra (on log-log axes) represents the level of nonstationarity or fluctuation behaviour. Depending

on the fluctuation behaviour, the magnitude and the sign of slopes can differ on each day. Positive slopes for lowest scales





indicate that the radiation is a nonstationary pure power law process or the fractional differenced process (e.g. Percival and Walden, 2005). These slope values contain the information of the fluctuation behaviour for the whole day, and can form as a basis for any cluster analysis with the radiation datasets. For all the classified days into different predominant sky conditions (see table 1), positive slopes are observed with a marked difference in their magnitudes, which might be associated with the day-to-day fluctuation behaviour. The overcast (1.0–1.1) and broken cloudy (0.9–1.2) days are highly nonstationary in comparision to clear (0.5–0.7) and overcast (0.5–0.63) days. A detailed discussion on the relation between cloud structure and radiation variabilities will be subject of future work.

Considering the cumulative explained variance, it can be seen that for broken clouds, high-frequency variability contributes most strongly to the total variance of the global transmittance (Fig. 5b). For other sky conditions (overcast, cirrus and clear), only a small decrease in variability ($\sim 10\,\mathrm{W\,m^{-2}}$) is observed, if the averaging period is increased from $1\,\mathrm{min}$ up to $3\,\mathrm{h}$. In case of broken clouds, the corresponding decrease is about three times ($\sim 34\,\mathrm{W\,m^{-2}}$) the value observed for other sky conditions.

## 4.2 Spatial autocorrelation

An important aspect for assessing the density of a measurement network is the representativeness of observations at one station for other close-by network stations as a function of their distance. To investigate this aspect for the network operated during the HOPE campaign, the spatial autocorrelation $\rho$ has been determined as a function of station distance for the wavelet smooth $S_3$ and the wavelet details $D_3$ to $D_9$ of global transmittance, and are shown in Fig. 6. In this plot, points represent the correlation coefficient obtained for the individual station pairs. Results are again shown separately for different sky conditions. The autocorrelation is generally found to decrease as station distance and frequency increases, with significant differences notable depending on sky conditions.

The behavior of the spatial autocorrelation ($\rho$) as a function of distance between stations ($d$ in $\mathrm{km}$) is shown as blue line in Fig. 6, and has been modeled by an exponential decay function as given below:

$$\rho(d) = exp\left[-\left(\frac{d}{a}\right)^b\right] \tag{3}$$

Here, $a$ (in $\mathrm{km}$) and $b$ (dimensionless exponent) represent fit coefficients. If the station distance is negligible ($d \to 0$), then $\rho \to 1$ (perfect correlation). Similarly, if the station distance is infinite (i.e., $d \to \infty$), then $\rho \to 0$ (no correlation). We have applied the Levenberg-Marquardt least-squares fitting technique to determine the fit coefficients. When the correlation drops below the e-folding value (i.e., $\rho \leq \frac{1}{e}$), the associated distance between stations is defined as the decorrelation length. This occurs when the fit coefficient $a$ equals to the station distance $d$ and thus $a$ is referred as the decorrelation length. Figure 7 confirms our expectation that the decorrelation length decreases for increasing frequencies, following an approximately linear trend with slightly different slopes and offsets depending on sky conditions. The quality of fit, as measured by the root mean square error ($rmse$) between the observed and modeled correlations, has been found to increase linearly with decreasing frequency.

An overview of various parameterizations used for modeling the behavior of the spatial autocorrelation function as a function of station distance is presented in Table 3. Long and Ackerman (1995) used a linear model to parameterize the dependence



of correlation of the time series of global radiation measurements at different sites based on their distance of separation ($\lesssim 100$ km). Subsequently, the same linear function was used to fit the correlation of wavelet smooths corresponding to the transmittance (from Multi-Filter Rotating Shadowband Radiometer) and reflectance (from Meteosat SEVIRI pixels) as a function of distance in the study by Deneke et al. (2009). They observed that the correlation falls off faster than linear at small
distances due to the exponent ($c < 1$). In a study on the correlation between the solar power generation of solar cell inverters, it was shown that the correlation was dependent on the distance between the inverters, the wavelet time scales and the amplitude of daily fluctuations (Perpiñán et al., 2013). They used an exponential decay model with some constraints on the fit coefficients. The spatial decorrelation of the time series of SEVIRI pixels for its solar and infrared channels was studied for different cloud amounts as a function of distance ($\lesssim 200$ km) at different locations over Europe (Slobodda et al., 2015). In the most recent
study, Lohmann et al. (2016) used the Hoff and Perez (2012) model of spatial correlation using a range of cloud speeds from 2 to $10\,\mathrm{m\,s^{-1}}$, and demonstrated that Hoff and Perez (2012) model is not able to capture the correlation structure for mixed sky conditions.

In our study, the spatial correlation of transmittance variations decays faster than linear at small distances as is indicated by the exponent ($b$) of Eq. 3, and depends strongly on the type and/or amount of clouds. Small-scale cloud features significantly
decrease the correlation on days with broken clouds. We point out a likely influence of the cloud speed on the decorrelation length. Additionally, anisotropy in the decorrelation relative to the direction of cloud motion is expected, and might influence the observed relationship (Hinkelman, 2013). In the following parts of the paper, the empirically fitted autocorrelation functions are used to represent the spatial variability at a given temporal frequency across the observation domain.

### 4.3 Spatial representativeness of a point measurement

The spatial representativeness of a point measurement at the center of a domain of interest depends on the size of the domain, the temporal averaging applied, and the spatio-temporal variability present in the observations. Generally, higher variability leads to a reduction of representativeness. Statistically optimum methods for spatial averaging have been developed to provide spatial means including uncertainty estimates when using data from a number of stations (Kagan, 1979; Gandin, 1993), and allow us to provide an estimate of the representativeness error, defined here as deviation of point measurement from spatial mean
for a considered domain. These techniques have been previously applied to global surface air temperature and precipitation measurements, and surface networks of soil moisture observations (Vinnikov et al., 1990, 1999). Similarly, the observations from our high density pyranometer network can be used to evaluate or quantify the uncertainties due to small-scale cloud inhomogeneity during validation studies. In this paper, we assume that the global transmittance field within the observation domain is statistically homogeneous and isotropic. A concise mathematical treatment for quantifying the effects of spatial and
temporal averaging is given in Appendix A.

In Fig. 8, the power spectra of area-averaged transmittances for different domain sizes are compared to those of a point measurement. They generally follow a similar trend compared to the point measurement irrespective of sky conditions, but show a stronger decrease of variability with increasing frequency and area, which illustrates that spatial averaging acts as lowpass filter. At lower frequencies corresponding to time periods of 1.5—3 h, only minor differences in variability are notable for time





series for spatially extended domains and point observations, at least for the range of domain sizes considered here. However, at higher frequencies, the variability of the spatially-averaged global transmittance is significantly reduced compared to that of the point observation, irrespective of the prevailing sky condition. Again, the variance of spatially-averaged transmittance is observed to be higher under broken clouds at all frequencies and spatial resolutions, compared to that for clear, cirrus and overcast conditions.

At $10 \times 10 \, \mathrm{km}^2$ and for variations corresponding to time periods of 1.5–3 h, the spatially-averaged variance is lower by 10% (clear), 16% (cirrus), 18% (overcast) and 38% (broken clouds) than the variability observed by a single station. Even for a domain size of $1 \times 1 \, \mathrm{km}^2$, the spatially-averaged variance is 2–4% lower than the variability obtained by a point measurement for the considered sky conditions.

The level of similarity between two time series is often expressed by metrics such as the explained variance or the root mean square error, and suitable averaging time scales are often determined by studying the sensitivity of these metrics to the choice of averaging scale. The explained variance ($\gamma_D^2$), given by the square of the cross-correlation between the time series at a single station and its area-averaged counterpart is used here and shown in Fig. 9 for different sky conditions, including its dependence on the temporal scales of averaging for the three domain sizes. The explained variance is thus used here as a measure to quantify the synchronicity of variations, while the power spectrum quantifies their mean amplitude. An exponential decay of the explained variance is observed as the temporal frequency increases for all spatial domain sizes. As expected, the deviation between a single station and an area average becomes larger at higher frequencies and for larger spatial areas. In consequence, the variations observed at a single station should no longer be used to predict the variations of the area-averaged transmittance at higher frequencies and for larger domains. Further, the explained variance between the wavelet smooth $S_3$ (3 h) of the point measurement and the area-averaged values of global transmittance is insensitive to the domain sizes. The decorrelation times for which the point and area-averaged variations become essentially uncorrelated is defined here by the e-folding value of $e^{-1}$ (= 0.368) for the correlation, and are listed in table 4 for the different sky conditions.

Finally, the deviation between point observations and spatial averages is determined for different domain sizes and temporal averaging periods, combining the two effects discussed before. The magnitude of the expected deviation as a function of domain size and temporal averaging period is shown in Fig. 10 for different sky conditions. It is generally observed that the representativeness error increases with the size of the spatial domain, and decreases for longer averaging periods. Also, the error converges against a limit value at high frequencies, indicating that the contribution of high-frequency variability beyond the frequency range considered here only causes a negligible further increase of the representativeness error.

Table 5 provides a quantitative estimate of the deviations of both global transmittance and corresponding global radiation for 3 different domain sizes, 3 averaging periods, and for different sky conditions. As the averaging frequency interval and domain size increases, the deviation between point measurement and corresponding area averages increases, irrespective of the prevailing sky condition. As expected, the range of deviations for both long (3 h, $S_3$) and short (5.25 s, including $D_{13}$) averaging periods is largest for broken clouds.





On clear days, the representativenss error of a point measurement for an area-averaged mean value increases only slightly as the averaging period decreases, and ranges from 2.1 % to 3.3 %. The difference between the maximum and minimum deviations resulting from the choice of averaging period is found to be around 0.6 % ($\sim 4\,\mathrm{W\,m^{-2}}$) regardless of spatial domain.

The range of deviations of a point observation under cirrus clouds is found to be around 1.6 % ($\sim 11\,\mathrm{W\,m^{-2}}$ for a $1{\times}1\,\mathrm{km^2}$) domain, and 2 % ($\sim 14\,\mathrm{W\,m^{-2}}$ for both $3.163{\times}3.163\,\mathrm{km^2}$ and $10{\times}10\,\mathrm{km^2}$) domains, and for 3h temporal averaging. A strongly increasing linear trend of the deviations is found from a reduction of averaging period, indicating that small-scale changes of cirrus cloud properties resulting from microphysical, dynamical and radiative processes can be removed effectively by sufficiently long temporal averaging. Dobbie and Jonas (2001) investigated the structure and lifetime of cirrus clouds using model simulations, and conclude that radiation together with latent heating leads to much more dynamic and inhomogeneous clouds.

During overcast skies, the representativeness error again increases substantially with increasing domain size, doubling and tripling its magnitude when going from $1{\times}1\,\mathrm{km^2}$ to domain sizes of $3.163{\times}3.163\,\mathrm{km^2}$ and $10{\times}10\,\mathrm{km^2}$ for short averaging periods. While the small deviations for a $1{\times}1\,\mathrm{km^2}$ domain below 1 % indicate that the hemispheric nature of a pyranometer measurement is able to resolve variability at the kilometer scale well, large-scale variations in cloud optical properties lead to deviations up to 3.3 % in transmittance or $22.6\,\mathrm{W\,m^{-2}}$ for a $10{\times}10\,\mathrm{km^2}$ domain.

As expected, the magnitude of deviations in global transmittance and corresponding radiation due to the limited representativeness of a point observation is found to be distinctly higher for all considered domain sizes and frequency intervals under broken cloudy situations. It varies from 4.5–11.5 % ($\sim 31.1$–$78.3\,\mathrm{W\,m^{-2}}$) over spatial areas ranging from $1{\times}1$ to $10{\times}10\,\mathrm{km^2}$. Again, deviations decrease strongly with increasing averaging period by more than 50 % for 3 h averaging. An interesting observation is that the representativeness error at different spatial resolutions seems to converge for 1 h or longer averaging periods. Hinkelman et al. (2007) reported that cumulus cloud inhomogeneity gave rise to an instantaneous error in global radiation of up to $40\,\mathrm{W\,m^{-2}}$ or even higher at different solar zenith angles. This well-known "broken-cloud effect" arises from variability in the direct and diffuse radiation (based on solar position), and can lead to an enhancement of global radiation above clear-sky conditions. As a result, large inconsistencies can occur for collocated satellite and surface measurements during broken cloudy conditions. Similarly, Barker and Li (1997) reported signficant deviations from 1D radiative transfer due to horizontal photon transport if the horizontal dimensions of a considered atmospheric column are decreased. Horvath and Davies (2004) provided further evidence for the relevance of 3D radiative effects through the observed anisotropy in the reflected solar radiation, which increasingly deviates from 1D radiative transfer if the spatial resolution of the satellite is increased. Zinner and Mayer (2006) reported that at $1\,\mathrm{km}$ scale, the errors associated with horizontal photon transfer and the plane parallel approximation cancel at least to some degree for stratiform boundary layer clouds

Based on our findings for different sky conditions, the comparison of time series corresponding to spatial averages of global radiation on the one hand, and point measurements on the other hand, can result in large deviations due to the limited representativeness of the point measurement. Similar effects are expected occur for other observables such as liquid water path. To address this issue of representativeness, we recommend here to apply a low-pass filter which removes variability at higher frequencies without significant correlation. Even for lower frequencies, a low-pass filter should be applied to adjust the power





spectrum of a point time series towards that of the spatially averaged time series, at least if the reduction factor of the amplitude of variations shown in Fig. 5 can be estimated. Nevertheless, significant deviations cannot always be avoided, but should be quantified, for example using the methodology introduced in this paper.

## 5   Summary and Conclusions

A unique dataset of global radiation observations has been collected using a dense network of pyranometer stations (Madhavan et al., 2016) during the HOPE Jülich and Melpitz Column (MCOL) campaigns (Macke and HOPE-Team, 2016), and is analyzed in this paper to characterize the small-scale spatio-temporal variability of the global radiation field. The individual time series have been subjected to a multiresolution analysis based on the Haar wavelet following the methodology of (Deneke et al., 2009). Characteristic properties have been identified from this analysis for clear sky, cirrus, broken cloud and overcast conditions. Power spectra for the individual time series and the spatial autocorrelation function are presented. A method has been introduced to assess the representativeness of the time series of a point measurement compared to results for a larger area centered around the measurement location. This method allows to determine the optimal accuracy that can be achieved for the validation of satellite products for a given pixel footprint, or the evaluation of an atmospheric model with a given grid-cell resolution.

The most significant findings of this study are summarized as follows:

i. The power spectra of global transmittance exhibit unique characteristics for different prevailing sky conditions associated with the dominant cloud type. The power spectra are mainly determined by the power spectra of the direct-beam transmittance. This behavior is observed even in overcast conditions, likely due to remaining periods with significant direct irradiance. For days with broken clouds, the variability of global transmittance is significantly and distinctly higher for all considered frequencies than for other situations, and contains remarkable contributions ($1\% \sim 7\,\mathrm{W}\mathrm{m}^{-2}$) even at high frequencies below $1\,\mathrm{min}$. This finding is noteworthy as a recommendation for the operation of BSRN network stations, which only require to store $1\,\mathrm{min}$ averages (McArthur, 2005), thereby missing significant amounts of variability

ii. The spatial autocorrelation between stations decreases strongly with increasing frequency. Variations at different points separated by more than $1\,\mathrm{km}$ are completely uncorrelated for higher frequencies above $3\,\mathrm{min}$. The decorrelation lengths decrease linearly with increasing frequency (on a log-log scale) and a distinct dependence on cloud and sky conditions was not observed (see Fig. 7).

iii. While the time series of spatially averaged irradiance fields generally resemble the behavior of a point measurement, its power spectrum is strongly attenuated (96-98% for $10\times10\,\mathrm{km}^2$, 80-90% for $3.163\times3.163\,\mathrm{km}^2$, 55-80% for $1\times1\,\mathrm{km}^2$) at higher frequencies ($\sim 1\,\mathrm{min}$) and for larger domains. Variations between the spatial average and the point measurement are not correlated at high frequencies. As a consequence, only a small fraction of the high-frequency variability within an extended domain can be explained by a point measurement




iv. As a consequence of the previous conclusions, point measurements can deviate strongly from the spatial mean of a surrounding domain. This effect can reach as much as $80\,\mathrm{W\,m^{-2}}$ for a grid-box of $10\times10\,\mathrm{km^2}$ during broken cloud conditions. For a comparison of time series of a point measurement with that of a spatially averaged value, the power spectrum of the point measurement should be adjusted to match that of the spatial average to ensure best correspondence. A low-pass filter should be applied to remove high frequencies for which the correlation drops below a certain threshold. To determine this threshold, the autocorrelation function has to be known, which however depends on the prevailing sky condition.

The methods presented in this paper allow for an explicit treatment of the effects of temporal and spatial averaging on the spatio-temporal variability of global radiation, and can easily be adapted to other geophysical fields. We have applied this methodology to estimate the inherent uncertainty arising from a comparison of two time series with fundamentally different spatial and temporal averaging scales, as is commonly done in radiation closure studies, the evaluation of atmospheric models or satellite products with point measurements. The findings contribute towards a better understanding of the uncertainties in such comparisons.

In future work, we plan to apply these findings towards an assessment of the level of accuracy of satellite-based estimates of shortwave irradiance from Meteosat SEVIRI with ground-based measurements (e.g. Deneke et al., 2008; Greuell et al., 2013), to separate retrieval uncertainties from the inherent uncertainty arising from the limited representativeness of one dataset for the other. Based on the results presented here, it is important to explicitly take into account the sky condition including their occurrence frequencies in the validation, as the representativeness error is situation-dependent and will therefore influence the validation statistics. A classification of sky conditions based on the observed power spectrum seems promising given the distinct features described above. However, the dependence of power spectra on cloud cover and solar elevation warrants further investigation. Finally, the pyranometer network observations include temperature measurements, allowing to study the correlation of variability in irradiance and temperature.

Due to the spatially distributed nature of the pyranometer network, the present work can also be extended to estimate Lagrangian instead of the Eulerian decorrelation scales, by considering the maxima in the time-lagged cross-correlation of transmittance time series observed at different sites. This time shift can be converted into an estimate of wind speed and direction, and will allow a separation of changes in radiation resulting from advective changes in clouds, which depend on the wind flow, and from temporal changes in cloud properties, which are independent of current wind speed and direction. Such an analysis will also enable a comparison of spatial and temporal decorrelation scales obtained from geostationary satellite observations (Bley et al., 2016).

Finally, this work can serve as reference for evaluating the representation of clouds including their radiative effects and spatial variability in high-resolution atmospheric models (Heinze et al., 2016), and thereby can contribute towards improved climate predictions. The spatial and temporal scaling properties of atmospheric transmittance are closely linked to those of the cloud fields. Significant deviations of modeled and observed values can thus be attributed to deficiencies in the simulation of clouds and their interaction with solar radiation (e.g. Harshvardhan et al., 1989; Pincus et al., 2008). Towards this goal, it is also important to clarify to what extent 3D radiative effects contribute to such deviations. A radiation closure study using



reconstructed 3D cloud distributions based on observations (see (e.g. Fielding et al., 2013)) as input to a radiative transfer code (e.g. Macke et al., 1999; Barlakas et al., 2016) could be an essential step towards this, and is planned for the future.

**Appendix A: Spatial representativeness of a point time series**

Let $\Psi(\boldsymbol{x}, t)$ represent the time series of a point measurement at point $\boldsymbol{x}$ in the observation domain of interest. The following statistical parameters are defined for this time series:

(i) The mean of the time series at $\boldsymbol{x}$ is given by:

$$\overline{\Psi(\boldsymbol{x}, t)} = \mathbb{E}\big[\Psi(\boldsymbol{x}, t)\big] \tag{A1}$$

(ii) The variance of the time series at $\boldsymbol{x}$ is given by:

$$var(\Psi(\boldsymbol{x}, t)) = \mathbb{E}\Big[\big(\Psi(\boldsymbol{x}, t) - \overline{\Psi(\boldsymbol{x}, t)}\big)^2\Big] \tag{A2}$$

(iii) The covariance of any two time series at $\boldsymbol{x_i}$ and $\boldsymbol{x_j}$ is given by:

$$cov(\Psi(\boldsymbol{x_i}, t), \Psi(\boldsymbol{x_j}, t)) = \mathbb{E}\Big[\big(\Psi(\boldsymbol{x_i}, t) - \overline{\Psi(\boldsymbol{x_i}, t)}\big) \cdot \big(\Psi(\boldsymbol{x_j}, t) - \overline{\Psi(\boldsymbol{x_j}, t)}\big)\Big] \tag{A3}$$

(iv) The autocorrelation $\rho$ between any two time series at $\boldsymbol{x_i}$ and $\boldsymbol{x_j}$ is given by:

$$\rho(\Psi(\boldsymbol{x_i}, t), \Psi(\boldsymbol{x_j}, t)) = \frac{cov(\Psi(\boldsymbol{x_i}, t), \Psi(\boldsymbol{x_j}, t))}{\sqrt{var(\Psi(\boldsymbol{x_i}, t) \cdot var(\Psi(\boldsymbol{x_j}, t)}} \tag{A4}$$

We now assume that the measurement field within the observation domain is statistically homogeneous (i.e., invariant under translation due to the shift in the origin of the coordinate system) and isotropic (i.e., invariant under rotations and reflections of the coordinate system). Consequently, the following properties hold:

(a) Homogeneity:

$\overline{\Psi(\boldsymbol{x}, t)} = \overline{\Psi}$ for all $\boldsymbol{x}$ and $t$, and

$var(\Psi(\boldsymbol{x}, t)) = C$ (i.e., with $C$ constant for all $\boldsymbol{x}$ and $t$.

(b) Isotropy:

$cov(\Psi(\boldsymbol{x_i}, t), \Psi(\boldsymbol{x_j}, t)) = f(d(\boldsymbol{x_i}, \boldsymbol{x_j}))$, where $d$ is the distance between the stations, and $f$ is a positive function defined for $d > 0$.

By adopting the above assumptions in Eq. A4, the autocorrelation $\rho$ becomes:

$$\begin{aligned}
\rho(\Psi(\boldsymbol{x_i}, t), \Psi(\boldsymbol{x_j}, t)) &= \frac{\mathbb{E}\big[\big(\Psi(\boldsymbol{x_i}, t) - \overline{\Psi}\big) \cdot \big(\Psi(\boldsymbol{x_j}, t) - \overline{\Psi}\big)\big]}{\mathbb{E}\big[\big(\Psi - \overline{\Psi}\big)^2\big]} \\
&= \frac{cov(\Psi(\boldsymbol{x_i}, t), \Psi(\boldsymbol{x_j}, t))}{var(\Psi)} \\
&= \rho(d(\boldsymbol{x_i}, \boldsymbol{x_j}))
\end{aligned} \tag{A5}$$





Therefore, the autocorrelation $\rho$ is a function of the distance $d$ between $\boldsymbol{x_i}$ and $\boldsymbol{x_j}$.

For a spatial area $A$, the area-averaged time series is obtained as:

$$\Psi_A(t) = \frac{1}{A} \iint\limits_A \Psi(\boldsymbol{x},t)d\boldsymbol{x} \tag{A6}$$

The following statistical parameters are found for the area-averaged time series:

5    (i) The mean of the area-averaged time series is given by:

$$\overline{\Psi_A(t)} = \mathbb{E}\big[\Psi_A(t)\big] = \overline{\Psi} \tag{A7}$$

(ii) The variance of the area-averaged time series is given by:

$$
\begin{aligned}
var(\Psi_A) &= \mathbb{E}\big[(\Psi_A - \overline{\Psi})^2\big] \\
&= \mathbb{E}\left[\frac{1}{A}\left(\iint\limits_A (\Psi(\boldsymbol{x_i},t) - \overline{\Psi})d\boldsymbol{x_i}\right) \cdot \frac{1}{A}\left(\iint\limits_A (\Psi(\boldsymbol{x_j},t) - \overline{\Psi})d\boldsymbol{x_j}\right)\right] \\
&= \frac{1}{A^2} \cdot \iint\limits_A \iint\limits_A \mathbb{E}\left[(\Psi(\boldsymbol{x_i},t) - \overline{\Psi}) \cdot (\Psi(\boldsymbol{x_j},t) - \overline{\Psi})\right]d\boldsymbol{x_i}d\boldsymbol{x_j} \\
&= \frac{1}{A^2} \cdot \iint\limits_A \iint\limits_A cov(\Psi(\boldsymbol{x_i},t), \Psi(\boldsymbol{x_j},t))d\boldsymbol{x_i}d\boldsymbol{x_j} \\
&= var(\Psi) \cdot \left[\frac{1}{A^2} \cdot \iint\limits_A \iint\limits_A \rho(d(\boldsymbol{x_i},\boldsymbol{x_j}))\,d\boldsymbol{x_i}d\boldsymbol{x_j}\right]
\end{aligned} \tag{A8}
$$

So, the variance of area-averaged time series is directly proportional to the variance of the time series centered in the
10    observation domain and the domain weighted autocorrelation function $\rho$.

Now, the statistical parameters between the time series centered in the domain and the area-averaged time series for the domain area $A$ are given below:

(i) The covariance of the time series $\Psi$ and the area-averaged value $\Psi_A$ is given by:

$$
\begin{aligned}
cov(\Psi, \Psi_A) &= \mathbb{E}\big[(\Psi - \overline{\Psi}) \cdot (\Psi_A - \overline{\Psi})\big] \\
&= \mathbb{E}\left[(\Psi(\boldsymbol{x},t) - \overline{\Psi}) \cdot \frac{1}{A}\left(\iint\limits_A (\Psi(\boldsymbol{x_i},t) - \overline{\Psi})d\boldsymbol{x_i}\right)\right] \\
&= \frac{1}{A} \cdot \iint\limits_A \mathbb{E}\big[(\Psi(\boldsymbol{x},t) - \overline{\Psi}) \cdot (\Psi(\boldsymbol{x_i},t) - \overline{\Psi})\big]d\boldsymbol{x_i} \\
&= \frac{1}{A} \cdot \iint\limits_A cov(\Psi(\boldsymbol{x},t), \Psi(\boldsymbol{x_i},t)d\boldsymbol{x_i} \\
&= var(\Psi) \cdot \left[\frac{1}{A} \cdot \iint\limits_A \rho(d(\boldsymbol{x},\boldsymbol{x_i}))d\boldsymbol{x_i}\right]
\end{aligned} \tag{A9}
$$





(ii) The square of the cross-correlation $\gamma_A$ (or explained variance) of the time series centered in the observation domain $\Psi$ and the area-averaged value $\Psi_A$ is obtained as the ratio of the square of the corresponding covariance to the product of their individual variances (using Eqs. A8 and A9).

$$\begin{aligned}
\gamma_A^2 &= \frac{\left[cov(\Psi, \Psi_A)\right]^2}{var(\Psi) \cdot var(\Psi_A)} \\
&= \frac{\left[\iint_A \rho(d(\boldsymbol{x}, \boldsymbol{x_i})) d\boldsymbol{x_i}\right]^2}{\left[\iint_A \iint_A \rho(d(\boldsymbol{x_i}, \boldsymbol{x_j})) d\boldsymbol{x_i} d\boldsymbol{x_j}\right]}
\end{aligned} \tag{A10}$$

In order to quantify the variance of the difference between the time series $\Psi$ and the area-averaged value $\Psi_A$, we assume that the variance of the area-averaged time series is linearly related to the variance of the point time series with an optimal filter, $\alpha_A$ (see Eq. A8) defined as below:

$$var(\Psi_A) = \alpha_A \cdot var(\Psi) \tag{A11}$$

     Now, the variance of the difference between the point time series $Psi$ and the area-averaged time series $\Psi_A$ (i.e., the
unexplained variance) is given by:

$$\begin{aligned}
var(\Psi - \Psi_A) &= var(\Psi_A) + var(\Psi) - 2 \cdot cov(\Psi_A, \Psi) \\
&= var(\Psi_A) + var(\Psi) - 2 \cdot \gamma_A \cdot \sqrt{var(\Psi_A) \cdot var(\Psi)} \\
&= (\alpha_A + 1) \cdot var(\Psi) - 2 \cdot \gamma_A \cdot \sqrt{\alpha_A} \cdot var(\Psi) \\
&= \left[\alpha_A + 1 - 2 \cdot \gamma_A \cdot \sqrt{\alpha_A}\right] \cdot var(\Psi)
\end{aligned} \tag{A12}$$

Expressing Eq. A12 in terms of the standard deviation, the area-averaging error $\delta$ between the point time series $\Psi$ and area-averaged time series $\Psi_A$ can be obtained as below:

$$\delta(\Psi - \Psi_A) = \sqrt{(\alpha_A + 1 - 2 \cdot \gamma_A \cdot \sqrt{\alpha_A})} \cdot \delta(\Psi) \tag{A13}$$

Alternately, we define a damped time series $\Psi'$ as the representative variability at a single station and as given below:

$$\Psi' = \sqrt{\alpha_A} \cdot (\Psi - \overline{\Psi}) + \overline{\Psi} \tag{A14}$$

The above Eq. A14 implies that:

$$var(\Psi') = \alpha_A \cdot var(\Psi) \tag{A15}$$

     The variance of the difference between the point time series $\Psi'$ and the area-averaged time series $\Psi_A$ is then given by:

$$\begin{aligned}
var(\Psi' - \Psi_A) &= var(\Psi_A) + var(\Psi') - 2 \cdot cov(\Psi_A, \Psi') \\
&= \alpha_A \cdot var(\Psi) + \alpha_A \cdot var(\Psi) - 2 \cdot \gamma_A \sqrt{var(\Psi_A) \cdot var(\Psi')} \\
&= 2 \cdot \alpha_A \cdot var(\Psi) - 2 \cdot \gamma_A \sqrt{\alpha_A \cdot var(\Psi) \cdot var(\Psi')} \\
&= 2 \cdot \alpha_A \cdot (1 - \gamma_A) \cdot var(\Psi)
\end{aligned} \tag{A16}$$





Expressing the Eq. A16 in terms of the standard deviation, the area-averaging error $\delta$ between the point time series $\Psi$ and area-averaged time series $\Psi_A$ can be obtained as below:

$$\delta(\Psi' - \Psi_A) = \sqrt{2 \cdot \alpha_A - 2 \cdot \alpha_A \cdot \gamma_A} \cdot \delta(\Psi) \tag{A17}$$

Comparing Eqs. A13 and A17, we find $\delta(\Psi' - \Psi_A) < \delta(\Psi - \Psi_A)$ as $\gamma_A \leq 1$

5   *Acknowledgements.* The authors acknowledge essential technical support from the TROPOS mechanics and electronics workshops in designing and building the autonomous pyranometer, especially Cornelia Kurze and Hartmut Haudek. Many thanks to all the private landowners for their support. We are grateful to the Research Center Jülich (FZJ) for their valuable logistic support in setting up and maintaining the instruments. The first author acknowledges the funding support of the Federal Ministry of Education and Research (BMBF), Germany as part of the $HD(CP)^2$ project (Grant No. 01LK1212C). We thank our colleague Fabian Senf, and Piet Stammes from KNMI,
10  the Netherlands for their valuable comments and suggestions during various stages of this manuscript. We also thank John Kalisch, Sebastian Bley, Daniel Merk, Felix Dietzsch, Timo Hanschmann, Michael Eickmeier, Anja Hünerbein, Felix Peintnet, Alexander Graf (FZJ), and Ronny Badeke for extending their support during the HOPE campaigns. Daily TSI time-azimuth data was provided by Dr. Jan H. Schween from the group of Prof. Dr. Susanne Crewell at the University of Cologne, Köln. We appreciate the support provided by Mr. Alexander Los and Mr. Kees Hogendijk from EKO Instruments, the Netherlands.





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





**Table 1.** Classification of days into clear, cirrus, overcast, and broken cloudy sky condition during the HOPE Jülich and Melpitz-Column campaigns conducted in the years 2013 and 2015.

| Sky condition | Observation days (day/month) | |
| --- | --- | --- |
| | HOPE (2013) | MCOL (2015) |
| Clear | 04/05, 08/06, 09/07, 21/07 | 04/07 |
| Cirrus | 22/04, 24/04, 16/07 | – |
| Overcast | 09/06, 28/06 | 21/06 |
| Broken clouds | 13/04, 25/04, 01/05, 02/05, 24/05, 04/06, 19/07 | 17/06 |

**Table 2.** Averaging time periods corresponding to each wavelet detail ($D_J$).

| Wavelet detail | $D_3$ | $D_4$ | $D_5$ | $D_6$ | $D_7$ | $D_8$ | $D_9$ | $D_{10}$ | $D_{11}$ | $D_{12}$ | $D_{13}$ |
| --- | --- | --- | --- | --- | --- | --- | --- | --- | --- | --- | --- |
| Time periods | 1.5 – 3.0 h | 45 – 90 min | 22.5 – 45 min | 11.25 – 22.5 min | 5.6 – 11.25 min | 2.8 – 5.6 min | 1.4 – 2.8 min | 42 – 84 s | 21 – 42 s | 10.5 – 21 s | 5.25 – 10.5 s |

**Table 3.** Summary of various parameterizations used for modeling the behavior of spatial autocorrelation $\rho$ as a function of the station distance $d$.

| Literature reference | Parameterization | Remarks |
| --- | --- | --- |
| Long and Ackerman (1995) Deneke et al. (2009) | $\rho = a - b \cdot d^c$ | $a$, $b$ and $c$ are fit coefficients |
| Hoff and Perez (2012) Lohmann et al. (2016) | $\rho = \left(1 + \dfrac{d}{\Delta t \cdot \Delta v}\right)^{-1}$ | $\Delta t$ is the time interval, and $\Delta v$ is the relative cloud speed |
| Perpiñán et al. (2013) | $\rho = a + b \cdot exp\left(-\dfrac{d}{c}\right)$ | $a$, $b$ and $c$ are fit coefficients |
| Slobodda et al. (2015) | $\rho = 1 - \dfrac{d^b}{a}$ | $a$ and $b$ are fit coefficients |
| Present study | $\rho = exp\left[-\left(\dfrac{d}{a}\right)^b\right]$ | $a$ and $b$ are fit coefficients |





**Table 4.** E-folding times (min) for the explained variance between the point measurement and area-averaged values under different sky conditions.

| Sky condition | A=1×1 km$^2$ | A=3.163×3.163 km$^2$ | A=10×10 km$^2$ |
|---|---|---|---|
| *Clear* | 6 min | 21 min | 49 min |
| *Cirrus* | 2 min | 7 min | 28 min |
| *Overcast* | 1 min | 4 min | 26 min |
| *Broken clouds* | 4 min | 15 min | 70 min |

**Table 5.** Mean deviation between point measurement and spatial averages of global transmittance ($\delta T$) and corresponding global radiation ($\delta G$, in W m$^{-2}$) for different averaging time periods and domain sizes.

| | | Averaging time periods | | | | | |
|---|---|---|---|---|---|---|---|
| **Sky** | **Domain** | $S_3$ (3.0 h) | | $D_6$ (10.25 min) | | $D_{13}$ (5.25 s) | |
| **condition** | **size (km$^2$)** | $\delta T$ | $\delta G$ (W m$^{-2}$) | $\delta T$ | $\delta G$ (W m$^{-2}$) | $\delta T$ | $\delta G$ (W m$^{-2}$) |
| **Clear** | **1×1** | 0.0019 | 1.293 | 0.0213 | 14.518 | 0.0265 | 18.003 |
| | **3.163×3.163** | 0.0024 | 1.633 | 0.0252 | 17.146 | 0.0300 | 20.415 |
| | **10×10** | 0.0027 | 1.837 | 0.0284 | 19.295 | 0.0328 | 22.350 |
| **Cirrus** | **1×1** | 0.0046 | 3.130 | 0.0171 | 11.615 | 0.0287 | 19.513 |
| | **3.163×3.163** | 0.0082 | 5.579 | 0.0204 | 13.880 | 0.0335 | 22.789 |
| | **10×10** | 0.0101 | 6.872 | 0.0262 | 17.800 | 0.0375 | 25.547 |
| **Overcast** | **1×1** | 0.0015 | 1.021 | 0.0068 | 4.640 | 0.0092 | 6.260 |
| | **3.163×3.163** | 0.0033 | 2.245 | 0.0132 | 8.958 | 0.0184 | 12.519 |
| | **10×10** | 0.0074 | 5.035 | 0.0247 | 16.794 | 0.0333 | 22.657 |
| **Broken** | **1×1** | 0.0076 | 5.171 | 0.0468 | 31.844 | 0.0842 | 57.310 |
| **clouds** | **3.163×3.163** | 0.0148 | 10.070 | 0.0525 | 35.708 | 0.1024 | 69.673 |
| | **10×10** | 0.0178 | 12.111 | 0.0695 | 47.295 | 0.1151 | 78.288 |





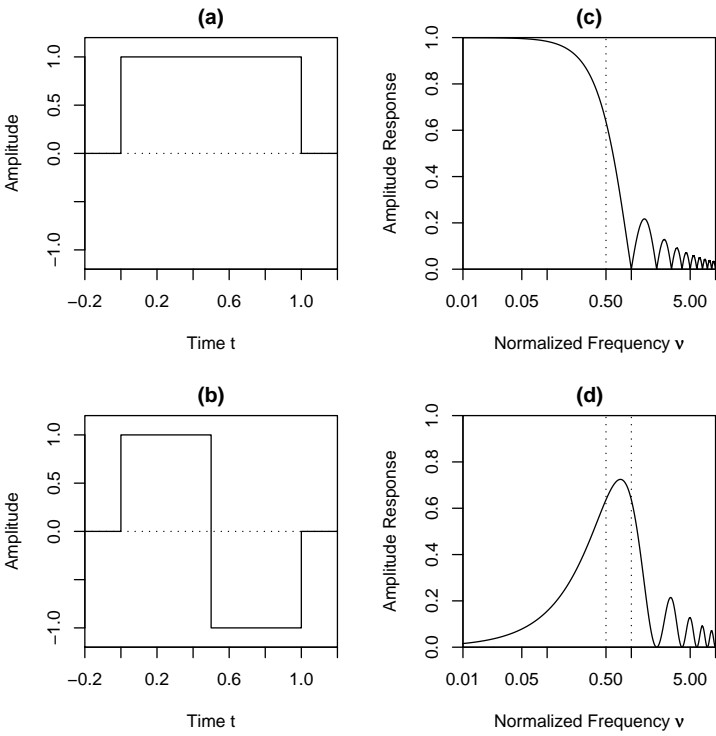

**Figure 1.** Time representation of the Haar (a) scaling and (b) wavelet function, and the frequency response of the associate (c) lowpass and (d) bandpass filters. Adopted from Deneke et al. (2009).





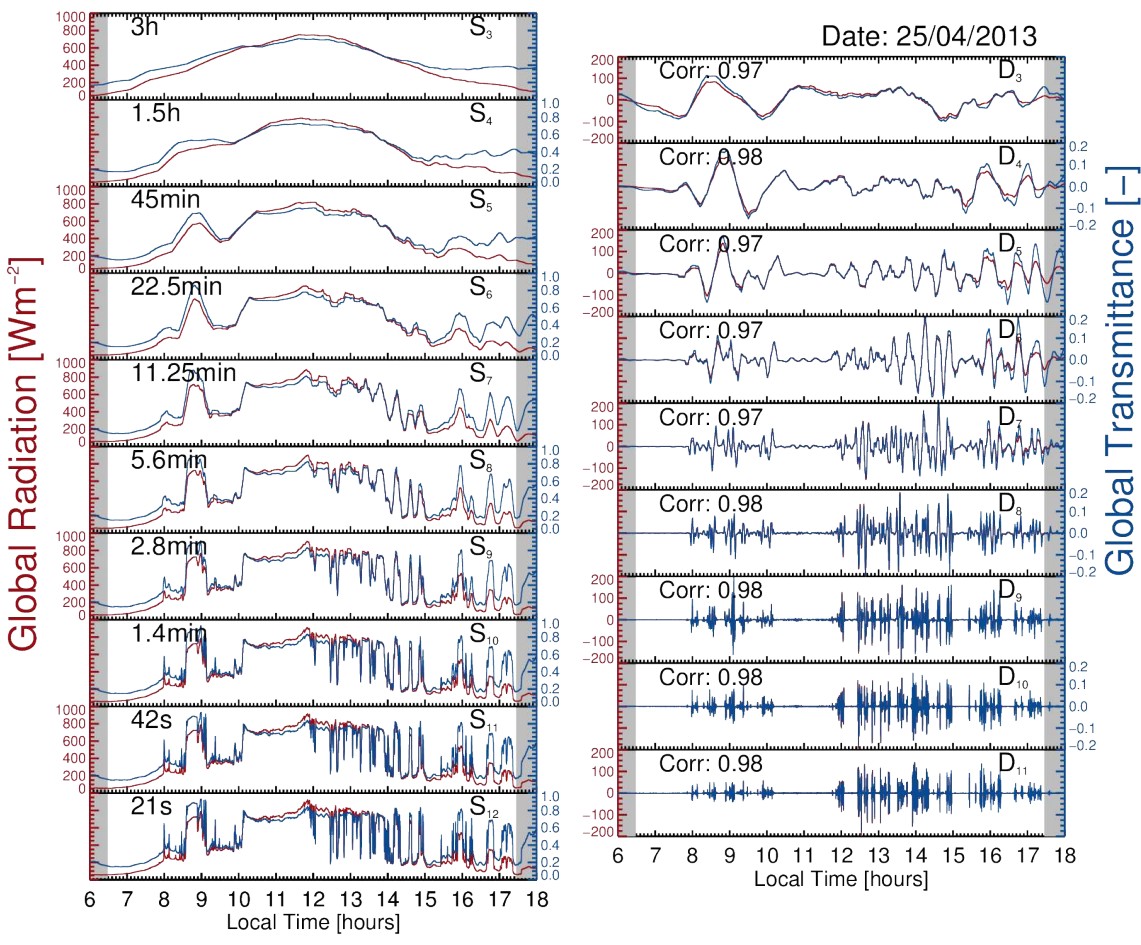

**Figure 2.** Multiresolution analysis of global radiation (red) and corresponding transmittance (blue) showing smooths (left panel) and details (right panel) as a function of local time (in h) for a pyranometer station at FZJ on 25 April 2013. Shaded gray region on both panels correspond to the region with solar zenith angle $> 75°$. The smoothing time is given in the left panels, while the correlation of the details in global radiation and transmittance is listed in the right panels.





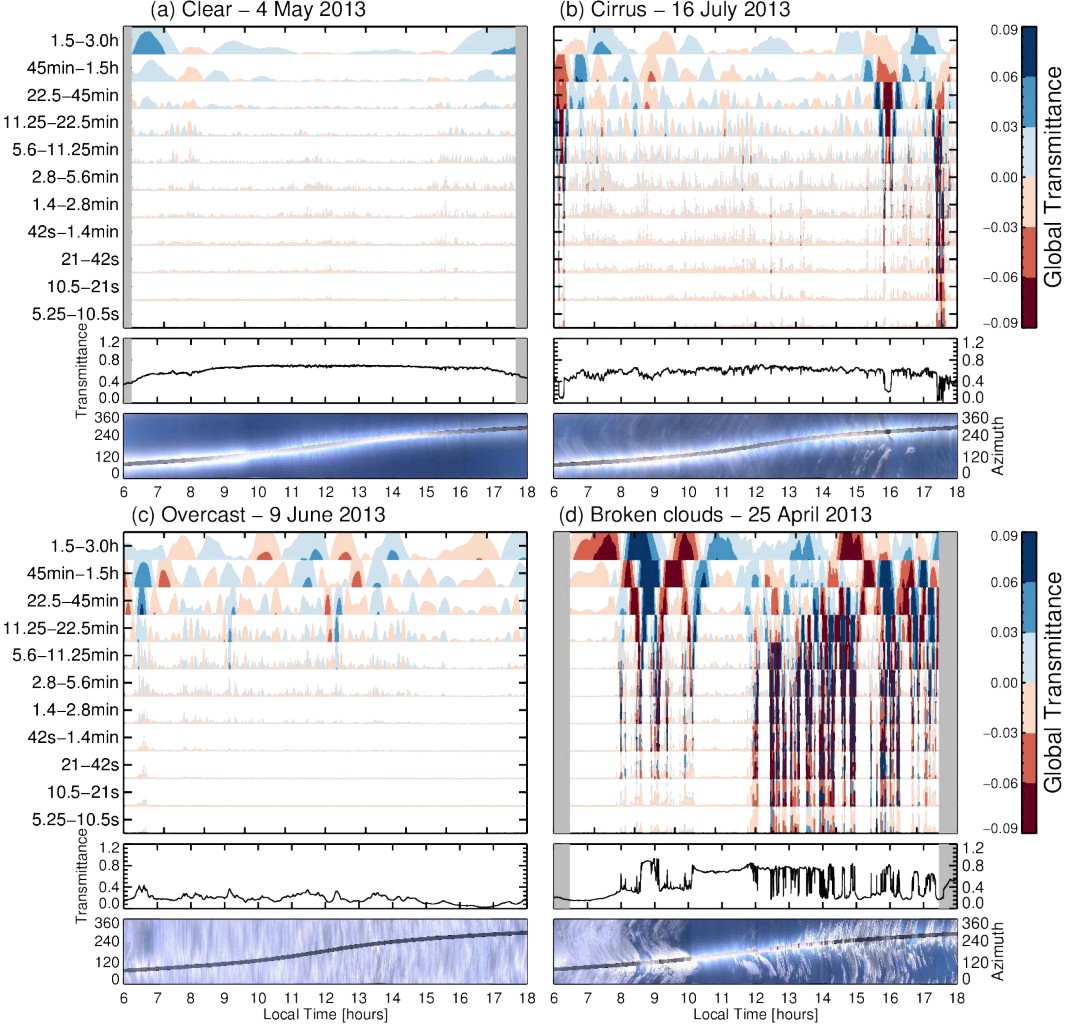

**Figure 3.** Horizon graphs for MRA of global transmittance from a pyranometer station at FZJ during the HOPE campaign represented as a function of local time (in h) for different sky conditions: (a) clear - 4 May 2013, (b) cirrus - 16 July 2013, (c) overcast - 9 June 2013, and (d) broken clouds - 25 April 2013. The top panels represent the horizon plots of transmittance details. Middle panels represent the original time series of transmittance. The $t - azi$ plots of the sky imager at $45°$ elevation angle are shown in the bottom panels. Shaded gray color in the top and middle panels of (a) and (b) correspond to the region with solar zenith angles $> 75°$. A horizon graph is constructed by dividing a normal line plot into bands defined by uniform value ranges. The bands are then layered to reduce the chart height. Negative values (red bands) can be mirrored or offset onto the same space as positive values (blue bands) such that the colors are differentiated. These layered bands are nested together. Such a visualization allows us to identify extraordinary behaviors or predominant patterns, view changes, interpret each of the time series independently from the others and perform comparisons between the different temporal periods (Few, 2008).





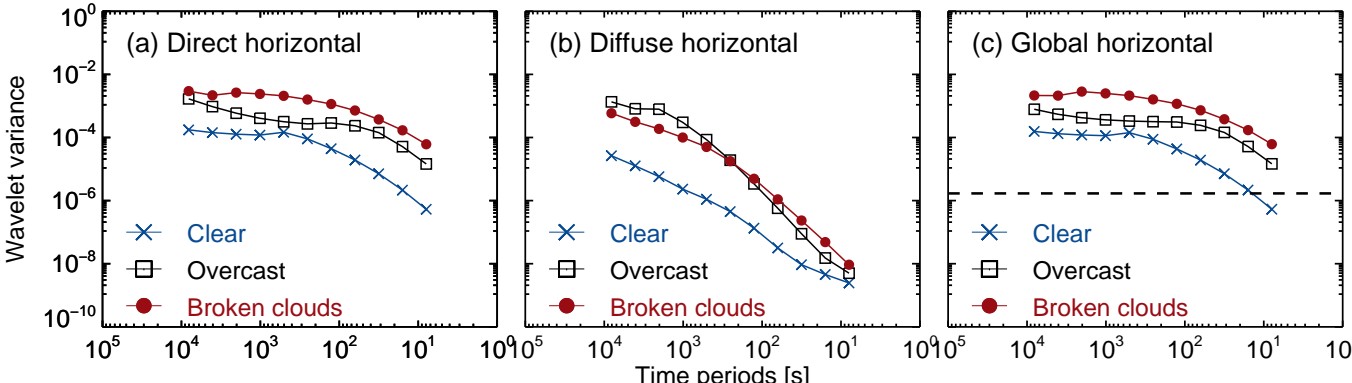

**Figure 4.** Power spectrum of the (a) direct, (b) diffuse, and (c) global transmittance as a function of temporal frequency for different sky conditions obtained during Melpitz Column (MCOL) experiment for days listed in Tab. 1. The black dashed horizontal line in (c) represents the combined measurement uncertainty of the pyranometer system.

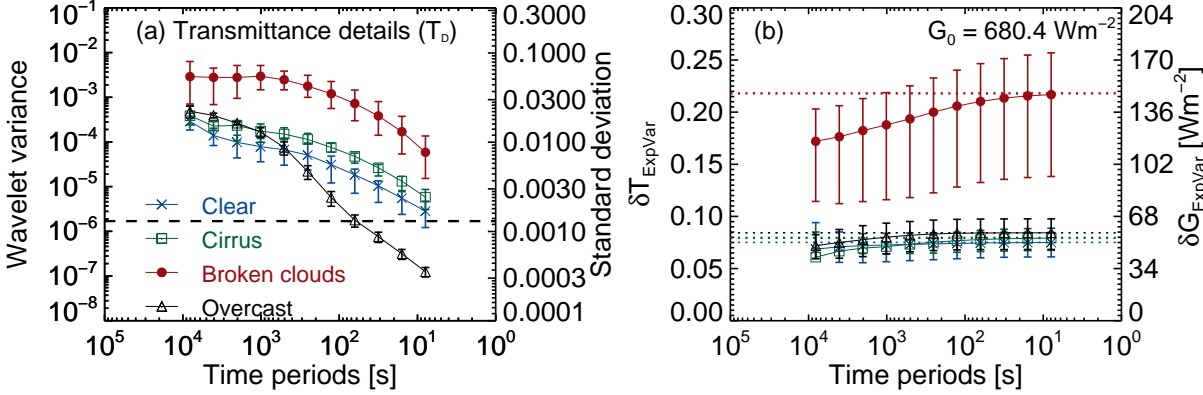

**Figure 5.** (a) Wavelet variance, and the (b) cumulative variance (from Eq. 1) of global transmittance from all the pyranometer stations in the observation domain as a function of considered frequency/averging period for cases during the HOPE campaign. The vertical bars around the mean value represent the observed minimum and maximum variances. The dashed horizontal lines in (b) denote the total variance of the original time series averaged across all stations within the observation domain.

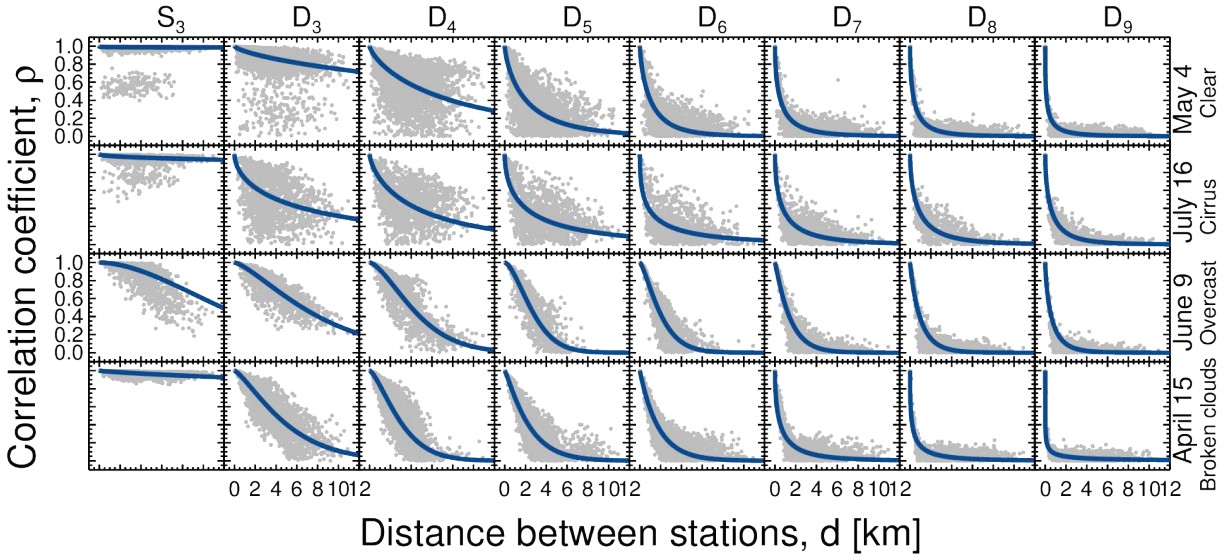

**Figure 6.** Spatial autocorrelation $\rho$ as a function of station distance $d$ for days with different sky conditions: (a) clear - 4 May 2013 (top row), (b) cirrus - 16 July 2013 (second row), (c) overcast - 9 June 2013 (third row), and (d) broken clouds - 25 April 2013 (last row). Here, $S_3$ corresponds to the wavelet smooth of global transmittance at 3 h averaging time scale, while $D_3$ to $D_9$ represent the wavelet details of global transmittance.

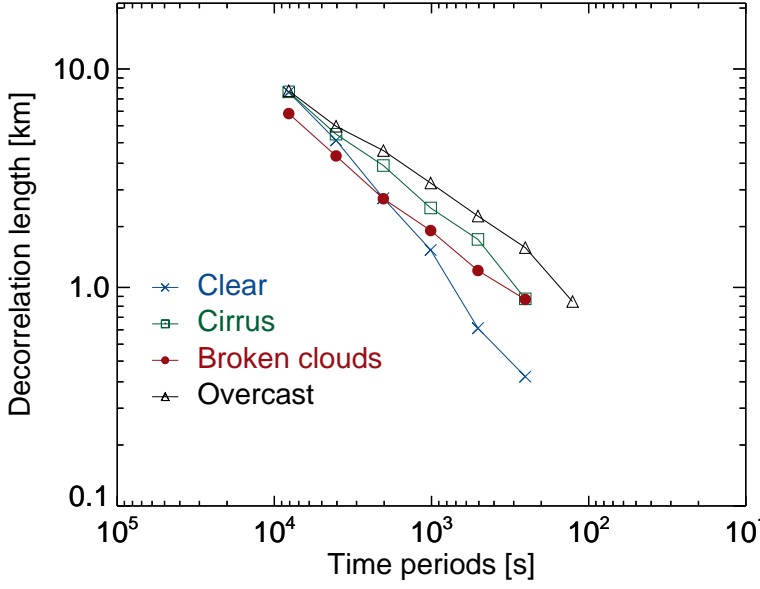

**Figure 7.** Decorrelation lengths $a$ (in km), determined as e-folding time of the spatial correlation function, and its dependence on the time period of variations.





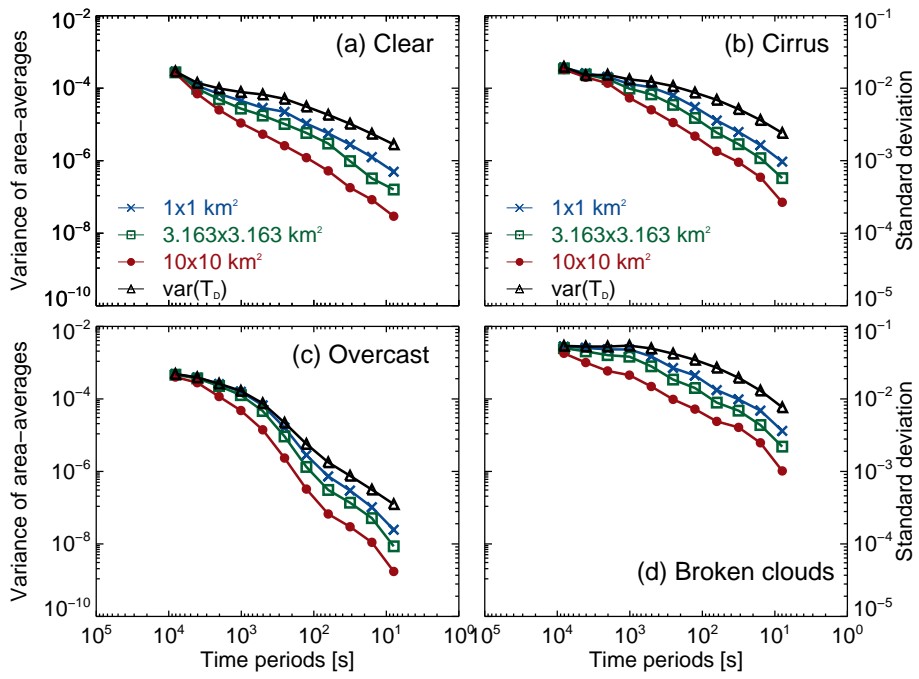

**Figure 8.** Power spectrum of spatially averaged transmittance as a function of frequency under different sky conditions: (a) clear, (b) cirrus, (c) overcast, and (d) broken clouds. $var(T_D)$ denotes the power spectrum of a point measurement of global transmittance as shown in figure 5a).





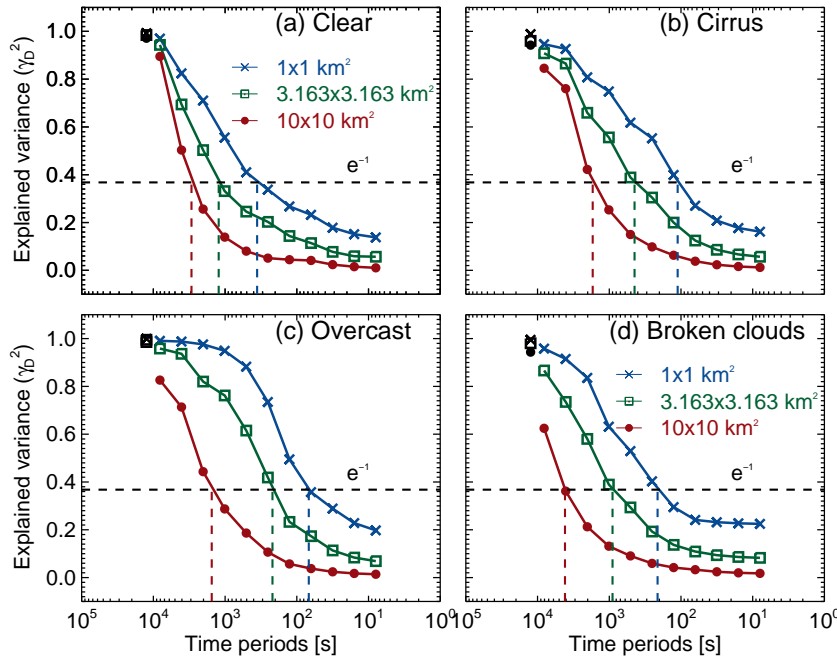

**Figure 9.** Explained variance ($\gamma_D^2$, the square of the cross-correlation) between the wavelet details of the point measurement and the area-averaged values of global transmittance as a function of their time period and for different domain sizes and sky conditions: (a) clear, (b) cirrus, (c) overcast, and (d) broken clouds. The black symbols respectively denote the explained variance ($\gamma_{S_3}^2$) of wavelet smooths $S_3$ (3h) corresponding to the domain size. The black dashed horizontal line in each of the sub-figures represent the e-folding time of $e^{-1} = 0.368$ and the dashed vertical lines corresponds to the decorrelation period for the selected domain sizes.





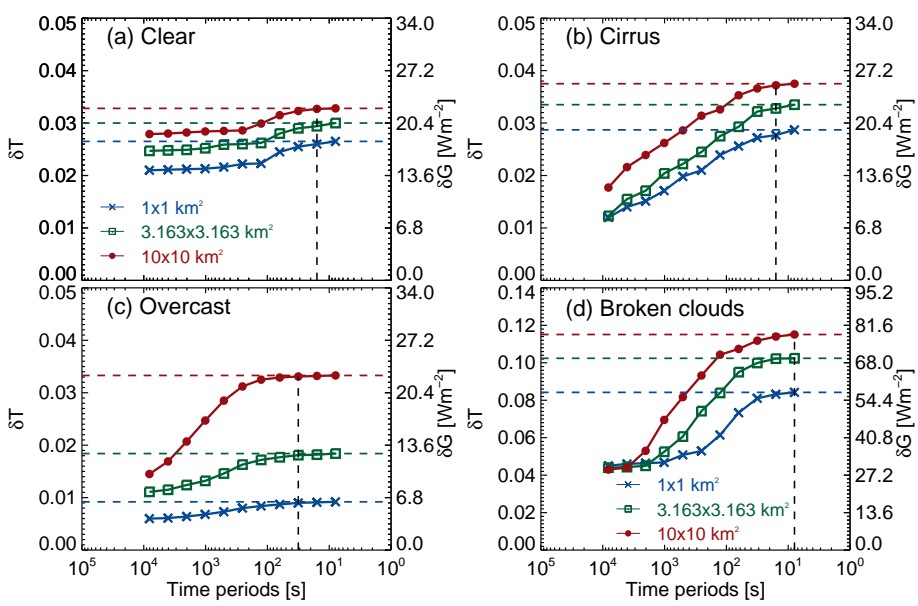

**Figure 10.** Area averaging error in the global transmittance ($\delta T$) and corresponding global radiation ($\delta G$, in $\mathrm{W\,m^{-2}}$) with different sky conditions: (a) clear, (b) cirrus, (c) overcast, and (d) broken clouds for different domain sizes represented as a function of averaging time periods. The dashed horizontal lines correspond to the maximum deviation observed for the different domain sizes. Dashed vertical lines represent the minimum averaging time above which the area averaging errors are less sensitive at different spatial resolutions.