# Peer review of "Multiresolution analysis of the spatiotemporal variability in global radiation observed by a dense network of 99 pyranometers"

_Atmospheric Chemistry and Physics, 2016_

## Referee Comment (RC1) · M. Nunez (Referee) · 30 Nov 2016

**Comments on *Multiresolution analysis of the spatiotemporal variability in global radiation observed by a dense network of 99 pyranometers during the HOPE campaign* by B.L. Madhavan et al. submitted for publication to *Atmospheric Chemistry and Physics***

Solar radiation at the surface varies considerably in time and space resulting from different factors including solar angle, atmospheric turbidity and surface albedo. At the regional/local scale extending from 0 to 10 km, clouds and other atmospheric constituents exert a strong influence on variability, raising questions on the representativeness of point measurements of solar radiation and the density of pyranometer networks that are needed for proper sampling. This variability also impacts on satellite estimates of solar radiation, since the technique usually relies on surface pyranometer measurements for validation. It is within this context that the authors have produced a thorough and timely study involving 99 pyranometers deployed in a 10 x12 km$^2$ area in Germany. A total of 19 days have been selected encompassing a range of cloud types and the data has been analysed using wavelet analysis at different spatial and temporal scales. In my view the analysis approach is appropriate, the quality of the work is good and the work makes a significant contribution to the literature.

It is good to note that the theory of wavelet analysis has been simplified in this revised version to make the document more readable to a general audience not familiar with the technique. I also note that more information on the pyranometer network is provided. I have some added comments that will hopefully help with the main focus of the study. I will organise my comments along the following topics:

- ***What is the analysis telling us regarding the contribution of different cloud spatial dimensions to the variability?***
  I am aware that a more detailed study is planned involving cloud interactions. However there is still need to link your results with some of the published literature. There are considerable number of studies who have described properties of stratocumulus clouds using power spectral analysis. They usually involve liquid water content, liquid water path or solar radiation transmission  with a power law relationship of the form:

$$Power(k) \approx k^{-C}$$

(a)

Where k is wave number ($2\pi/\lambda$, $\lambda$ being a distance unit in m or km) and C is a rregression constant. Studies involving aircraft, ground based, microwave radiometers and satellite data describe C as being around 5/3 but usually in a specific wavenumber range, equivalent to distances ranging from several tens pf km to less than 1 km (Boers, 1988; Cahalan and Snider, 1989;  Davis et al., 1999; Gerber et al., 2001).  Area distribution of broken Cu/Stcu clouds has  also been  studied Nunez et al. (2016), Koren et al., (2008),  and Cahalan and Snider(1989) which describe a typical distribution of cloud area A in terms of a given number density N :

$$\ln(N) = \ln(A)^{-C1} \qquad\qquad (b)$$

And C1 is a least square fit. These studies point to the importance of low wave numbers or large cloud areas in dominating the variance of the time series of liquid water and solar radiation transmission, with a partly cloudy scene dominated by a few large clouds and many smaller ones (eq. 2).

The authors should relate some of their results to the considerable published literature on the subject as listed above. For example, in Figure 2, large fluctuations are observed in details $D_3$, $D_4$ and $D_5$. These must be related to changes in transmission resulting from longer-term changes in dominant cloud structure and composition (S12 in Figure 2), or the equivalent of low wave numbers in equation (a). Higher number details do not show this as they examine local-scale variability in cloud features ($D_9$ to $D_{11}$).

In Figure 5(a), wavelet variance for all cloud conditions is largest at long time periods, implying that large cloud structures with their associated transmissions are important at this scale. Similarly the power spectrum in Figure 8 shows high variance at high time periods and the importance of large scale cloud structures. The authors should examine a least square fit for a single point measurement in Figure 8 within the larger context of power spectrum measurements (equation (a)). Transformation from period or frequency to wavenumber space may be accomplished using the frozen turbulence hypothesis (Cahalan and Snider (1989; p. 104)).

- ***Treatment of direct radiation***

Figure 4 shows that the power spectra of transmittance is determined by the power spectra of direct beam transmittance, which is also stated in the text at page 15, line 18. The statement is reasonable with cirrus, thin altostratus or partly cloudy liquid water clouds, but it is unlikely to hold for overcast liquid water clouds. Using a radiative model such as Libradtran-1.7 will show that direct irradiance is only around 4% of the global irradiance for liquid water clouds of optical depth 3. Given these conditions, it would be difficult to make a general statement that direct irradiance dominates the global irradiance spectrum. My advice is to restrict the study to liquid water clouds or provide a detailed cloud breakdown in Table 1 and state that the spectral results for direct irradiance refers to the specific set of conditions used.

- ***Treatment of clear skies***

It is interesting to see that the clear case in Figure 5(a) also exhibit a similar distribution with high variance at long time periods. It is unclear to me why this should be. Would aerosols and water vapour exhibit the same behaviour as clouds with regards to their transmission spectrum, with high variance at high time periods? Or perhaps it might be an artefact of the transmission calculation (G/G0, G is measured clear sky global radiation, G0 is the extra-terrestrial radiation)? At high zenith angles transmissions would be low due to higher air mass, imposing a strong diurnal change in clear sky transmission.

Table 4 shows that the variance between an point measurement and 1 km x 1 km average (wavelet smooth S3?) is uncorrelated after six minutes (decorrelation time) for clear conditions. In my opinion, this is a remarkably low figure. Again as in the above paragraph, what features of clear sky turbidity or instrumental errors are causing this behaviour?

- ***How widely applicablke are the results?***

The sections on autocorrelation and spatial representativeness are very good and should provide useful data when planning a pyranometer array. However the authors should provide a word of caution, probably in the Conclusion, that conditions sampled are typical of mid-latitude systems and that the results may not be applicable to other regions such as the tropics typified by local convection, large cumulonimbus clouds and weaker regional winds.

- ***Minor corrections***

Page 2, line 7: replace "up" by "updrafts"

Page 2, line 24: replace "…could show that especially…" by "…reported that spatially…".

Page 6, line 20: replace "… zenith angle below 75°" by "…zenith angle above 75°". Is this correct?

Page 8, line 21: replace "…wavelet-based spectra…" by "…wavelet-based spectral power density…".

Page 8, line 22: delete " The quality of fit…been found to increase linearly with decreasing frequency" to "The root mean square error (rmse) which measures the quality of fit has been found to decrease linearly with decreasing frequency".

Page 11, line 25: Side reflection from clouds is strongly enhanced in broken cloud conditions and could be important in lowering the correlation (Nunez et al., 2016).

Page 26, Table 3. It might be appropriate in the table to include averaging period used in the various studies (10 minutes, hourly, daily, etc.)

- ***References***

Boers, R., J. D. Spinhirne, and W. D. Hart, Lidar observations of the fine-scale variability of marine stratocumulus clouds, *J. Appl. Met.*, *27*, 797–810, 1988.

Cahalan, R. F., and J. B. Snider, Marine stratocumulus structure, *Remote Sens. Environ.*, *28*, 95–107, 1989.

Davis, A., A. Marshak, H. Gerber, and J. W. Wiscombe, 1999; Horizontal structure of marine boundary layer clouds from centimeter to kilometer scales, *J. Geophys. Res.*, *104*, 6123– 6144

Gerber, H., J.B. Jensen, A.B. Davis, A. Marshak, W.J. Wiscombe, 2001: Spectral density of cloud liquid water content at high frequencies, *J. Atmos. Sci.*, 497-503.

Koren, I., L. Oreopoulos, G. Feingold, L. A. Remer, and O. Altaratz (2008), How small is a small cloud?, *Atmos. Chem. Phys*., 8, 3855–3864.

Nunez, M., M.J. Marin-Fernandez, D. Serrano, M. P. Utrillas, K. Fienberg, J.A. Martinez-Lozano, 2016: Sensitivity of UV enhancement to broken liquid water clouds: a Monte Carlo approach as applied to Valencia, Spain, *J. Geophys. Res, Atmos*., 121, 949‑964.

---

## Referee Comment (RC2) · Anonymous Referee #2 · 30 Nov 2016

This paper takes advantage of the dense network of pyranometers deployed during the HOPE campaign to characterize with high detail the spatio-temporal variability of solar radiation at several scales. Thus, the paper addresses relevant scientific matters which are within the scope of ACP. The paper uses relatively known tools (mainly wavelet analyses) to a new and comprehensive dataset. Substantial conclusions are reached, which are based on methods and assumptions that are in general clearly outlined. In my opinion, the authors give proper credit to related work and clearly indicate their own new/original contribution.

[Figure]

The title reflects the content of the paper, but I would suggest two minor changes. First, I would suggest adding "solar" when "global radiation" is mentioned, or changing the adjective "global" and use "solar" instead. For me it's more relevant let the reader know that the paper is about solar radiation, which is pretty obvious but not explicitly specified right now. Second, I think that "during the HOPE campaign" can be removed from the title, as it is not totally relevant (note that the authors do not mention the campaign in the abstract) and the reader does not need to know what the HOPE campaign was.

The abstract provide a complete summary of methods, results, and conclusions. However, I suggest mentioning in some way in the paragraph the geographical location of the measurements. In addition, the sentence "For frequencies below 1.0 min$-1$, variations in transmittance become completely uncorrelated already after several hundred meters" needs clarification. First, because I think it would be "above" and not "below" not to enter in contradiction with the previous sentence. Second, because I can't find this result in the text of the paper. The most similar thing that I can find is in section 5, point ii, where 3 minutes (and not 1 minute) and 1 km (and not several hundred meters) are mentioned.

In general, the paper is well structured and clear; the language fluent and precise; mathematical formulae and symbols correctly defined and used; and tables and, particularly, figures are of great quality. I do have however one general comment and several minor comments and technical corrections to be considered.

General comment.

I suggest removing from the paper the part related with the measurements at the MORDOR site, including Figure 4 and the corresponding comments. I would say that the paper will have exactly the same value if the authors remove these parts. From my point of view, including the information about MORDOR campaign is confusing: I didn't understand, when I read section 2, why a campaign that was performed two years later had something to do with the HOPE campaign. Then in section 4 I realized that data

from MORDOR was needed to generate Figure 4, which is commented in as short as 8 lines of page 8. Figure 4 is intended to justify the use of global radiation, but I think that it is unnecessary. In addition, I think it is somewhat misleading. Under truly "overcast" conditions, direct radiation is null, so the power spectrum of the direct radiation under these conditions should be very different (impossible to compute, in fact) and the diffuse and global radiation should be identical. I understand that the Figure is generated by using only one "overcast" day, which probably was overcast with very thin clouds. In fact, for clouds with optical depth as low as 5, direct beam is totally extinguished, so the day used by the authors is not, from my point of view, a good example of a typical overcast day. If, as I suggest, the authors get rid of this part, they should also eliminate part of conclusion 5.i and a sentence from the abstract. I insist that without this part, the paper keeps being worth of publication, and, in my opinion, more "round", clear, and consistent.

Minor comments and technical corrections.

- P2, l 14-16. I wouldn't say that the references given to support the validation of satellite retrievals are the best ones. You could mention instead (or in addition), among many others: Enriquez-Alonso et al (2016), Norris and Evan (2015), Stubenrauch et al (2013).

** Norris JR, Evan AT (2015) Empirical removal of artifacts from the ISCCP and PATMOS-x satellite cloud records. J Atmos Ocean Technol 32:691–702. doi:10.1175/JTECH-D-14-00058.1

** Stubenrauch CJ, Rossow WB, Kinne S et al (2013) Assessment of global cloud datasets from satellites: project and database initi- ated by the GEWEX Radiation Panel. Bull Am Meteorol Soc 94:1031–1049. doi:10.1175/BAMS-D-12-00117.1

**Enriquez-Alonso, A., A. Sanchez-Lorenzo, J. Calbó, J. A. González, and J. Norris, 2016: Cloud cover climatologies in the Mediterranean obtained from satellites, surface observations, reanalyses, and CMIP5 simulations: validation and future scenarios. Clim. Dyn., 47, 249–269, doi:10.1007/s00382-015-2834-4.

- P4, l28-29. The phrase "As the spectral composition of the measured global radiation in the field deviates due to non-uniform spectral sensitivity" is unclear to me, as it mixes what happens in the field with the instrument sensitivity. Or maybe the problem is that I don't understand from what the measured radiation "deviates"?

- Eq. (1), eq (2). If the right hand side depends of index J, the left hand side, var(T) should also come with this subindex, shouldn't it?

- P7, l 11 (and elsewhere) Why it is relevant that the second considered domain is 3.163 km? Shouldn't 3.2 km be accurate enough? Why do you use square domains instead of circular areas of a given radius from the central point?

- P7, l 17-19. The sentence is not clear. Maybe you could use some other equation to explain what are you doing here?

- P8, l 12-13. ". . ., obtained as average power spectrum obtained from all pyranometer stations" Do you mean that you first obtained a power spectrum for each station measurements, and then you averaged all these spectra?

- P8, l 20-23. At the end of the sentence, you could mention that this phenomenon is usually referred to as "enhancement effect".

- P9, eq (3) and related comments. I understand that all this development is for "details" D3-D9, and not for "smooth" S3. But you might consider making this explicit in the text.

- P10, l 28-30. Ok with this sentence, but it is unclear to me how do you actually compute the "area-averaged" transmittances. I assume that you use all measurements from all stations in a given domain. Could you mention how many sites where included in each domain besides the central station? I understand that you used only one 10x10 km2 domain (since this is the size of the whole HOPE campaign domain) but did you use one of more domains of the other sizes (1x1, 3.2x3.2 km2)?

- P13, l25. Besides what I already told above (regarding what it is written in the abstract) here a potential confusion is evident. You say "for higher frequencies above 3min", but minutes is a unit of time (period) and not of frequency. Here and elsewhere in the text, you should be cautious when mixing the use of frequency and time periods. In fact, this is also relevant regarding many figures, which are correctly labeled in "time periods", but using a reverse axis, and in general, commented in the text in the frequency domain. I suggest that at least in the first figure where time period is used, you highlight in the caption the use of the reverse axis and the relation with frequency.

- P13, l 31-32. "As a consequence, only a small fraction of the high-frequency variability within an extended domain can be explained by a point measurement" Isn't this the other way around? That is, only a small fraction of the high-frequency variability recorded in a point measurement can be described by area-averaged (satellite, model, reanalysis) data?

- P14, conclusion iv. You should add to what time average correspond the value you give here (80 W m-2)

- Could you please double-check or update the links to WMO documents? I checked both links and they didn't work for me.

- Table 5. I think that 4-5 significant figures for the delta-G values are not required (and in fact, do not make sense). Indeed, in the text you correctly work with 2-3 significant figures (e.g. 79 instead of 78.288).

- Caption fig. 3. Shaded gray color bands are found in panels (a) and (d) (not b).

---

## Author Comment (AC1) · 14 Feb 2017

We thank the reviewer (Dr Manuel Nunez) for providing his valuable comments and suggestions on our article "Multiresolution analysis of the spatiotemporal variability of global radiation observed by a dense network of 99 pyranometers during the HOPE campaign" (acp-2016-694). In the process of revision, we have made the following corrections in the original manuscript:

- Title of the manuscript has been revised as "*Multiresolution analysis of the spatiotemporal variability in global radiation observed by a dense network of 99 pyra-*

[Figure]

*nometers*" (based on the suggestion of referee# 2).

**Major comments**

• **What is the analysis telling us regarding the contribution of different cloud spatial dimensions to the variability?**

- We have examined the least square fitting for the single point measurements (Figure 5 is included as an additional figure in the revised manuscript) and the details of our findings are incorporated in the subsection 4.1 along with the in- clusion of relevant literature as suggested. However, it should be noted that the global irradiance is a hemispherically integrated property and thus there cannot be an exact one-to-one relation to the cloud variability or to (directional) radiance variability. Finding an appropriate smoothing kernel requires intensive investiga- tions of the interaction of clouds and radiation including 3D radiative effects, and is beyond the scope of this study.

- A new summary table (Table 3) outlining the spectral exponents and scale regimes from different studies has been included.

• **Treatment of direct radiation**

- We agree with that the transmittance in overcast scenes should not be explained by direct radiation.

- Fig. 4 (old manuscript) is Fig. 11 in the revised manuscript.

- The results of Fig. 4 (old manuscript) are quite relevant and important for under- standing the variability of global radiation, and how the direct/diffuse contributions affect overall variability. Instead of dropping Fig. 4 (old manuscript), we would like to keep it by moving it to the very end of the results section, and make clear that these results are more of an outlook to future research than final results. Hence,

the text from page 8, lines 3-10 are moved to a short subsection 4.4, which also makes it clear that this is only an initial assessment and clarifies that the large direct contribution in overcast situations is likely due to our not very strict classi-fication of situations. In particular, even on the days classified as overcast, some periods with significant direct irradiance due to cloud gaps were observed and evidently dominate the power spectrum of the global transmittance.

- **Treatment of clear skies**

- The following statement is included in the subsection 4.1 - "Due to the changes in solar elevation and thus airmass over the day, a pronounced diurnal cycle in global transmittance is observed in clear sky situations, which introduces signifi-cant variance at longer time periods."

- The following statements are included in the subsection 4.3 - "The e-folding time of 6 min indicates that variations with frequencies higher than $1/6$ $\text{min}^{-1}$ are more or less uncorrelated between the point measurement and a spatial area of $1\times1$ $\text{km}^2$. It should also be noted that the spatial average has a significantly lower power spectral density at these frequencies. We thus think these variations are thus associated with small-scale fluctuations in clear sky turbidity only evident in the point measurements, possibly induced by small scale structure in water vapor and/or aerosols. However, we cannot rule out that such variability corresponds to undetected small clouds or even measurement artefacts such as shading of the instruments by birds."

- **How widely applicable are the results?**

- Fully agreed! We have added a sentence to the conclusions that our study is representative for mid-latitude summer conditions.

**Minor corrections**
* * *
**Interactive
comment**

- **Page 2, line 7: replace "up" by "updrafts".**

- Corrected. Replaced "up" by "updraughts".

- **Page 2, line 24: replace "... could show that especially ..." by "... reported that spatially ...".**

- Corrected.

- **Page 6, line 20: replace "... zenith angle below 75$^o$" by "... zenith angle above 75$^o$". Is this correct?**

- This is not correct. We have obtained the MRA results for solar zenith angle below 75$^o$ to exclude edge effects.

- **Page 8, line 21: replace "... wavelet-based spectra ..." by "... wavelet-based spectral power density ...".**

- Corrected.

- **Page 8, line 22: delete "The quality of fit ... been found to increase linearly with decreasing frequency" to "The root mean square error ($rmse$) which measures the quality of fit has been found to decrease linearly with decreasing frequency".**

- The statement is corrected.

- **Page 11, line 25: Side reflection from clouds is strongly enhanced in broken cloud conditions and could be important in lowering the correlation (Nunez et al., 2016).**

- The statement is included in the revised manuscript.

- **Page 26, Table 3: It might be appropriate in the table to include averaging period used in the various studies (10 minutes, hourly, daily, etc.).**

- Table 3 in the old manuscript is Table 4 in the revised manuscript. Table 4 is revised as suggested.

---

## Author Comment (AC2) · 14 Feb 2017

We thank the reviewer for providing his/her valuable comments and suggestions on our article "Multiresolution analysis of the spatiotemporal variability of global radiation observed by a dense network of 99 pyranometers during the HOPE campaign" (acp-2016-694). In the process of revision, we have made the following changes in the original manuscript:

- **Title of the manuscript -** *The title reflects the content of the paper, but I would suggest two minor changes. First, I would suggest adding "solar" when "global*

[Figure]

*radiation" is mentioned, or changing the adjective "global" and use "solar" instead. For me, it's more relevant let the reader know that the paper is about solar radiation, which is pretty obvious but not explicitly specified right now. Second, I think that "during the HOPE campaign" can be removed from title, as it is not totally relevant (note that the authors do not mention the campaign in the abstract) and the reader does not need to know what the HOPE campaign was.*

- We decided to keep the term global radiation in the title, as the AMS Glossary defines it in a consistent meaning to our usage as: Solar radiation, direct and diffuse, received from a solid angle of $2\pi$ steradians on a horizontal surface, see http://glossary.ametsoc.org/wiki/Global_radiation.

- The title of the manuscript is revised as - "Multiresolution analysis of the spatiotemporal variability in global radiation observed by a dense network of 99 pyranometers".

• **Abstract -** *The abstract provide a complete summary of methods, results, and conclusions. However, I suggest mentioning in some way in the paragraph the geographical location of the measurements. In addition, the sentence "For frequencies below 1.0 min$^{-1}$, variations in transmittance become completely uncorrelated already after several hundred meters" needs clarification. First, because I think it would be "above" and not "below" not to enter in contradiction with the previous sentence. Second, because I can't find this result in the text of the paper. The most similar thing that I can find is in section %, point ii, where 3 minutes (and not 1 minute) and 1 km (and not several hundred meters) are mentioned.*

- The information related to the geographical location of the measurements is included in the abstract.

- The revised sentence is as follows: "For frequencies above 1/3 min$^{-1}$ and points separated by more than 1 km, variations in transmittance become completely

uncorrelated."

- **General comment -** *I suggest removing from the paper the part related with the measurements at the MORDOR site, including Figure 4 and the corresponding comments. I would say that the paper will have exactly the same value if the authors remove these parts. From my point of view, including the information about MORDOR campaign is confusing: I didn't understand, when I read section 2, why a campaign that was performed two years later had something to do with the HOPE campaign. Then in section 4, I realized that data from MORDOR was needed to generate Figure 4, which is commented in as short as 8 lines of page 8. Figure 4 is intended to justify the use of global radiation, but I think that it is unnecessary. In addition, I think it is somewhat misleading. Under truly "overcast" conditions, direct radiation is null, so the power spectrum of the direct radiation under these conditions should be identical. I understand that the Figure is generated by using only one "overcast" day, which probably was overcast with very thin clouds. In fact, for clouds with optical depth as low as 5, direct beam is totally extinguished, so the day used by the authors is not, from my point of view, a good example of a typical overcast day. If, as I suggest, the authors get rid of this part, they should also eliminate part of the conclusion 5.i and a sentence from the abstract. I insist that without this part, the paper keeps being worth of publication, and in my opinion, more "round", clear, and consistent.*

- While we understand the concerns related to the lengthy description of the MORDOR measurements, the results of Fig. 4 (old manuscript) are quite relevant and important for understanding the variability of global radiation, and how the direct/diffuse contributions affect overall variability. Instead of dropping Fig. 4 (old manuscript), we would like to keep it by moving it to the very end of the results section and make clear that these results are more of an outlook to future research than final results. Hence, the text from page 8, lines 3-10 are moved to a short subsection 4.4, which also makes it clear that this is only an initial assessment and clarifies that the large direct contribution in overcast situations is likely due to our not very strict classification of situations. In particular, even on the days classified as overcast, some periods with significant direct irradiance due to cloud gaps were observed and evidently dominate the power spectrum of the global transmittance.

**Minor comments and technical corrections:**

- **P2, L14-16:** *I wouldn't say that the references given to support the validation of satellite retrievals are the best ones. You could mention instead (or in addition), among many others: Enriquez-Alonso et al (2016), Norris and Evan (2015) and Stubenrauch et al (2013).*

- These references are included in the revised manuscript.

- **P4, L28-29:** *The phrase "As the spectral composition of the measured global radiation in the field deviates due to non-uniform spectral sensitivity" is unclear to me, as it mixes what happens in the field with the instrument sensitivity. Or maybe the problem is that I don't understand from what the measured radiation "deviates"?*

- The revised statement is as follows: "Changes in the spectral distribution of downward irradiance compared to the conditions during calibration can lead to errors of up to 5%, particularly at higher solar zenith angles."

- **Eq. (1), Eq (2):** *If the right-hand side depends on index J, the left-hand side var(T) should also come with this subindex, shouldn't it?*

- The equations (1) and (2) are corrected.

- **P7, L11 (and elsewhere):** *Why it is relevant that the second considered domain is 3.163 km? Shouldn't 3.2 km be accurate enough? Why do you use square domains instead of circular areas of a given radius from the central point?*

- To maintain consistency with satellite pixel representation, we have defined the square domains.

- The second considered domain is corrected as $3.2 \times 3.2$ km$^2$ instead of $3.163 \times 3.163$ km$^2$.

• **P7, L12-13:** *The sentence is not clear. Maybe you could use some other equation to explain what are you doing?*

- Reference to the equations from the Appendix A are included in the statements.

- The revisions are done as follows: "... and $\alpha_A$ (from Eq. A11) is a linear reduction factor relating the variance of the point measurement (from Eq. A2) to the variance of an area-averaged time series (from Eq. A8). The explained variance (i.e., $\gamma^2_{S,J}$ and $\gamma^2_{D,J}$ from Eq. A10) between the point and area-averaged values are obtained separately for transmittance smooths ($S_J$) and details ($D_J$) for the different spatial and temporal scales. Then, the expected deviation $\delta_J$ for each wavelet detail is calculated based on the explained variance and summed to yield an estimate of the total variance, accounting for a reduced temporal variability of the spatially-averaged transmittance by the reduction factor."

• **P8, L12-13:** *"..., obtained as average power spectrum obtained from all pyranometer stations". Do you mean that you first obtained a power spectrum for each station measurements, and then you averaged all these spectra?*

- Yes. For better clarity, the statement is revised as follows - "The average power spectrum is obtained by considering the power spectra of all the pyranometer stations."

• **P8, L20-23:** *At the end of the sentence, you could mention that this phenomenon is usually referred to as "enhancement effect".*

- We have included this in the revised manuscript.

- **P9, Eq. (3) and related comments:** *I understand that all this development is for "details" D3-D9, and not for "smooth" S3. But you might consider making this explicit in the text.*

- Eq. 3 is applicable to the "details" ($D_3$ to $D_9$) as well as to the "smooths" ($S_3$ or any other smooth with a different index). This is also shown in Figure 6. So, there is no requirement for making an explicit statement in the text.

- **P10, L28-30:** *Ok with this sentence, but it is unclear to me how do you actually compute the "area-averaged" transmittances. I assume that you use all measurements from all stations in a given domain. Could you mention how many sites were included in each domain besides the central station? I understand that you used only one 10x10 km$^2$ domain (since this is the size of the whole HOPE campaign domain) but did you use one of more domains of the other sizes (1 $times$ 1, 3.2 $\times$ 3.2 km$^2$)?*

- In this paper, we utilise the spatial auto-correlation functions determined in the previous section to calculate the power spectral density of spatial averages and the deviation of spatial averages from point measurements.Thereby, we avoid averaging of multiple stations to obtain an approximation of a spatial average but rely on the assumption that the global transmittance field within the observation domain is statistically homogeneous and isotropic, and that its auto-correlation function follows Eq. 4.

- **P13, L25:** *Besides what I already told above (regarding what it is written in the abstract) here a potential confusion is evident. You say "for higher frequencies above 3 min", but minutes is a unit of time (period) and not of frequency. Here and elsewhere in the text, you should be cautious when mixing the use of frequency and time periods. In fact, this is also relevant regarding many figures, which*

*are correctly labeled in "time periods", but using a reverse axis, and in general, commented in the text in the frequency domain, I suggest that at least the first figure where time period is used, you should highlight in caption the use of the reverse axis and the relation with frequency.*

- The units of frequency and time are checked in the manuscript and corrected.

- The following statement is included in the caption of Figure 4 - "As the time period is inversely proportional to the frequency, the time periods (on the x-axis) are represented in ascending order of frequency scales. "

• **P13, L31-32:** *"As a consequence, only a small fraction of the high-frequency variability within an extended domain can be explained by a point measurement". Isn't this recorded in a point measurement can be described by area-averaged (satellite, model, reanalysis) data?*

- The statement is corrected in the revised manuscript.

• **P14, confusion iv:** *You should add to what time average correspond the value you give here ($80\ W\,m^{-2}$).*

- The statement is revised as following - "This effect can reach as much as 80 $W\,m^{-2}$ for a grid-box of $10\times10\ km^2$ corresponding to an averaging time period of 5.25–10.5 s ($D_{13}$) during broken cloud conditions."

• *Could you please double-check or update the links to WMO documents? I checked both links and they didn't work for me.*

- The links are updated and working.

• **Table 5:** *I think that 4-5 significant figures for the delta-G values are not required (and in fact, do not make sense). Indeed, in the text you correctly work with 2-3 significant figures (e.g. 79 instead of 78.288).*

- The values of $\delta G$ in Table 6 (in revised manuscript) are rounded to 2 significant digits in the decimal place.

- **Caption fig 3:** *Shaded gray color bands are found in panels (a) and (d) (not b).*

- Corrected.